# The HSV-1 ICP22 protein selectively impairs histone repositioning upon Pol II transcription downstream of genes

Lara Djakovic [1,6], Thomas Hennig[1,6], Katharina Reinisch[2,6], Andrea Milić[1], Adam W. Whisnant [1], Katharina Wolf [1], Elena Weiß [2], Tobias Haas[1], Arnhild Grothey[1], Christopher S. Jürges [1], Michael Kluge[2], Elmar Wolf [3,4], Florian Erhard [1], Caroline C. Friedel [2] ✉ & Lars Dölken [1,5] ✉

*Herpes simplex virus 1* (*HSV-1*) infection and stress responses disrupt transcription termination by RNA Polymerase II (Pol II). In *HSV-1* infection, but not upon salt or heat stress, this is accompanied by a dramatic increase in chromatin accessibility downstream of genes. Here, we show that the *HSV-1* immediate-early protein ICP22 is both necessary and sufficient to induce downstream open chromatin regions (dOCRs) when transcription termination is disrupted by the viral ICP27 protein. This is accompanied by a marked ICP22-dependent loss of histones downstream of affected genes consistent with impaired histone repositioning in the wake of Pol II. Efficient knock-down of the ICP22-interacting histone chaperone FACT is not sufficient to induce dOCRs in ΔICP22 infection but increases dOCR induction in wild-type *HSV-1* infection. Interestingly, this is accompanied by a marked increase in chromatin accessibility within gene bodies. We propose a model in which allosteric changes in Pol II composition downstream of genes and ICP22-mediated interference with FACT activity explain the differential impairment of histone repositioning downstream of genes in the wake of Pol II in *HSV-1* infection.

Productive *Herpes Simplex Virus type 1* (*HSV-1*) infection induces a profound shut-off of host gene expression by targeting multiple steps of RNA metabolism[1]. This curtails antiviral host responses and facilitates efficient, productive infection[2]. Disruption of transcription termination (DoTT) of cellular genes significantly contributes to *HSV-1*-induced host cell shut-off[3]. DoTT commonly results in read-through transcription extending for tens to hundreds of thousands of nucleotides beyond poly(A) sites. Read-through transcription originating from disrupted transcription termination often extends into downstream genes (denoted as "read-in" transcription for these genes)[3]. The

viral immediate-early protein ICP27 plays a direct, bimodal role in *HSV-1*-induced DoTT[4]. On the one hand, ICP27 interacts with and disrupts the essential mRNA 3'-end processing factor CPSF, thereby inducing the assembly of a dead-end 3' processing complex that is unable to cleave mRNA 3' ends. On the other hand, ICP27 acts as a sequence-dependent activator of mRNA 3' processing by binding to viral (and some host) transcripts thereby restoring CPSF activity and 3' mRNA cleavage. Interestingly, poly(A) read-through transcription is accompanied by a dramatic increase in chromatin accessibility downstream of the affected poly(A) sites[5]. In uninfected cells, open chromatin is

[1]Institute for Virology and Immunobiology, Julius-Maximilians-University Würzburg, Versbacher Straße 7, 97078 Würzburg, Germany. [2]Institute of Informatics, Ludwig-Maximilians-Universität München, Amalienstr. 17, 80333 Munich, Germany. [3]Cancer Systems Biology Group, Theodor Boveri Institute, University of Würzburg, Am Hubland, 97074 Würzburg, Germany. [4]Mildred Scheel Early Career Center, University of Würzburg, Beethovenstraße 1A, 97080 Würzburg, Germany. [5]Helmholtz Institute for RNA-based Infection Research (HIRI), Helmholtz-Center for Infection Research (HZI), 97080 Würzburg, Germany. [6]These authors contributed equally: Lara Djakovic, Thomas Hennig, Katharina Reinisch. ✉e-mail: caroline.friedel@bio.ifi.lmu.de; lars.doelken@uni-wuerzburg.de

predominantly observed around gene promoters as well as in the gene bodies of highly expressed genes, but also in intergenic regions[6]. During *HSV-1* infection, chromatin accessibility selectively increases downstream of genes with strong read-through transcription indicative of impaired histone repositioning in the wake of RNA polymerase II (Pol II) downstream of affected poly(A) sites. Disrupted transcription termination with Pol II transcription extending far downstream of gene 3′ ends (DoGs) is also observed upon cellular stress responses[7,8], influenza A virus infection[9,10] and cancer[11]. However, downstream open chromatin regions (dOCRs) do not arise upon salt or heat stress[5], indicating that transcription read-through is either not sufficiently strong for dOCR induction in stress responses or that additional viral factors are involved.

During the process of transcription, Pol II removes and subsequently repositions histones to facilitate efficient transcription elongation while maintaining chromatin architecture. The two transcription elongation factors and histone chaperons SPT6 and FACT (comprised of SSRP1 and SPT16) play a key role in this process (reviewed in[12]). The histone-chaperoning activity of the FACT complex facilitates both the removal and reassembly of histones in the wake of actively transcribing Pol II. Distinct regions of FACT interact with both the H2A-H2B dimer and the (H3-H4)$_2$ tetramer and can promote displacement of H2A-H2B from nucleosomes[13]. In particular, FACT mediates both histone eviction and repositioning at the early stages of transcription elongation[14]. SPT6 directly interacts with histones and assembles nucleosomes in vitro[15]. This activity is required for the maintenance of a chromatin structure that prevents improper usage of cryptic promoter elements, suggesting that SPT6 also reassembles nucleosomes in the wake of Pol II[15,16]. Interestingly, the viral ICP22 protein interacts with the FACT complex and relocalizes it to viral replication compartments (RCs)[17]. ICP22 (encoded by the *HSV-1* U$_S$1 gene) is an important, but non-essential, multifunctional viral 63-kDa polypeptide conserved in *Alphaherpesviruses* that is expressed with immediate early kinetics. *HSV-1* mutants lacking ICP22 do not form plaques in skin and lung fibroblasts but replicate in Vero or BHK cells[18]. ICP22 features a structurally-conserved core domain flanked by poorly-conserved, intrinsically disordered regions that are extensively phosphorylated by the viral pU$_L$13 protein kinase and, to a lesser extent, by the pU$_S$3 protein kinase[19,20]. ICP22 is also phosphorylated by yet unidentified cellular kinases and nucleotidylated by casein kinase II[21,22]. It is thus very likely that ICP22 exerts multiple different functions throughout productive virus infection that are governed by post-translational modifications. Accordingly, ICP22 has been shown to interact with a variety of different cellular proteins including cyclin-dependent kinase 9 (CDK9; pTEFβ)[23–25], cell division cycle 25C (CDC25C)[26] as well as FACT[17,27]. Notably, ICP22 facilitates the recruitment of FACT and SPT6 to RCs and to the transcribing Pol II complex[17,27]. Very early during productive infection, ICP22 induces the formation of virus-induced chaperone-enriched (VICE) domains, which may aid the proper folding of newly synthesized viral proteins[28]. It may by itself also function as a virally encoded co-chaperone (J-protein/Hsp40) by interacting with heat shock protein 70 (Hsc70) to prevent aggregation of misfolded proteins[29]. During productive infection, ICP22 and the viral pU$_L$13 protein kinase mediate an intermediate phosphorylation of the carboxyl terminal domain (CTD) of Pol II, termed Pol II$_i$[11,12]. ICP22 also alters the activity of the cell cycle component CDK1 to enhance viral late (L) gene expression[13]. This effect of ICP22 on key cell cycle regulators may explain the cell type-dependent growth defect of ICP22 null mutants[30]. Finally, it also plays an important role in *HSV-1* nuclear egress by interacting with the viral nuclear egress complex (pU$_L$31 and pU$_L$34 proteins)[31].

Here, we show that the viral ICP22 protein is necessary to induce dOCRs in *HSV-1* infection. Furthermore, ectopic expression of ICP22 was sufficient for the induction of dOCRs upon disruption of transcription termination and poly(A) read-through transcription induced by ectopic expression of ICP27. Read-through transcription and induction of dOCRs downstream of genes were associated with a notable depletion of histones downstream of gene 3′ends. Efficient knock-down of the histone chaperone FACT not only enhanced dOCR induction but also increased chromatin accessibility within gene bodies in an ICP22-dependent manner. It thereby alleviated the selective, DoTT-dependent increase in chromatin accessibility downstream of genes. We propose a model in which functional inhibition of FACT by the viral ICP22 protein and allosteric changes in Pol II composition downstream of the poly(A) signal result in selectively impaired histone repositioning in the wake of Pol II downstream of genes.

## Results

### Viral late gene expression is not required for dOCR induction

To identify the viral gene(s) responsible for the induction of dOCRs during *HSV-1* infection, we first assessed whether viral genome replication and, thus, viral late gene expression was required for the induction of dOCRs. We infected primary human fibroblasts (HFFFs) with *HSV-1* strain 17 for 8 h or strain F for 8 and 12 h in the presence or absence of the viral DNA polymerase inhibitor phosphonoacetic acid (PAA) and performed ATAC-seq ($n = 2$). The length of dOCRs was quantified for 4162 protein-coding and lincRNA genes that exhibited no read-in transcription originating from the read-through transcription of an upstream gene. These genes were identified in our previous study comparing transcriptional regulation in *HSV-1* strain 17 and Δvhs infection[32]. We excluded genes with read-in transcription as read-in transcription can result in dOCRs from an upstream gene extending into downstream genes (e.g., dOCRs for *SRSF3* extending into downstream *CDKNA1* gene, Supplementary Fig. 1a), thus confounding dOCR analyses. No further selection regarding whether analyzed genes are susceptible to either read-through transcription or dOCR induction was performed at this stage. In uninfected cells, the general absence of downstream open chromatin is reflected by only short (if any) dOCRs for the vast majority of cellular genes (Fig. 1a). Both wild-type (WT) strains 17 and F induced dOCRs for several hundred genes. However, this was slightly less prominent for strain F (Fig. 1a, example in Supplementary Fig. 1a), consistent with a lower extent of read-through transcription upon infection with this strain[4]. Strikingly, inhibition of viral DNA replication by PAA substantially increased dOCR lengths for both virus strains, while PAA treatment had no effect on uninfected cells.

It should be noted that changes in chromatin accessibility during *HSV-1* infection are not limited to induction of downstream open chromatin but also include other changes, e.g., changes at promoters that are associated with alterations in Pol II promoter occupancy. Here, we only focus on the induction of open chromatin downstream of genes during *HSV-1* infection. PAA treatment of *HSV-1*-infected cells was particularly useful for the investigation of dOCRs as it resulted in a much higher percentage of cellular reads, i.e., reads aligning to the human genome, in the ATAC-seq data due to the inhibition of viral DNA replication (Supplementary Data 1). For WT strain 17 infection, <50% of ATAC-seq reads originated from the host genome without PAA treatment, in contrast to >95% when viral DNA replication was inhibited by PAA. To exclude that the increase in dOCR induction by PAA was simply due to a greater sequencing depth on the cellular genome, we performed down-sampling of the respective sequencing libraries so that all samples had approximately the same number of reads mapping to the human genome. Even after down-sampling, significantly longer dOCRs were observed upon PAA treatment for both strains (Supplementary Fig. 1d). We thus hypothesize that enhanced levels of dOCRs upon inhibition of viral DNA replication by PAA treatment were the result of the following biological and technical phenomena: First, smaller amounts of viral DNA sequester fewer Pol II molecules away from the cellular chromatin, which reduces virus-induced transcriptional host shut-off. This, in turn, leads to substantially higher host transcriptional activity both within and downstream of genes[33], which

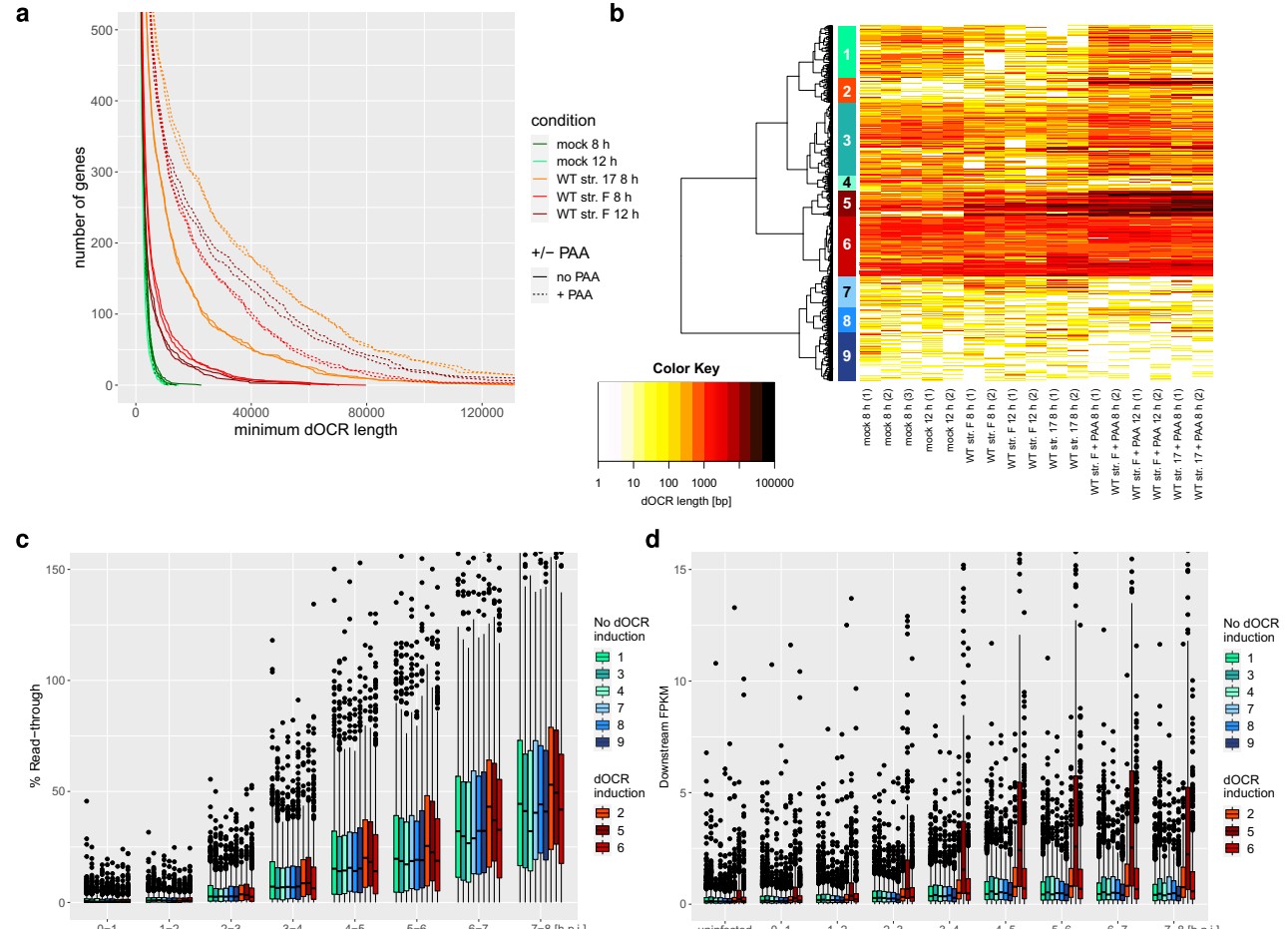

**Fig. 1 | Induction of dOCRs in HSV-1 infection is associated with downstream transcriptional activity. a** Number of genes with dOCR length greater than the value on the *x*-axis in mock and *HSV-1* WT strain 17 infections with or without PAA treatment (combined data of 2 biological replicates). To avoid having to define a threshold on whether a particular dOCR length is considered as dOCR induction, we visualized dOCR lengths in each condition for all analyzed 4162 genes without read-in transcription in *HSV-1* infection (excluding those with a dOCR length = 0). This depicts whether the number of genes with longer dOCRs is generally increased in the respective condition. The *y*-axis was limited to 500 to highlight differences in the number of genes with long dOCRs between mock and *HSV-1* infection. **b** Hierarchical clustering analysis (Euclidean distances, Ward's clustering criterion) of log10(dOCR length) for all analyzed genes (i.e., 4162 genes without read-in transcription in *HSV-1* infection) of the samples shown in (**a**). To define clusters, the cutoff on the clustering dendrogram was chosen such that three groups of genes

visually identified as showing dOCR induction in the heatmap resulted in separate clusters. Identified clusters are numbered from top to bottom as indicated and marked by colored rectangles. Shades of red indicate clusters with dOCR induction and shades of blue clusters without dOCR induction. **c**, **d** Boxplots showing the distribution of read-through transcription (**c**) and downstream FPKM (**d**) for the 9 clusters (*n* = 609, 290, 851, 176, 305, 701, 367, 289, and 574 genes for clusters 1–9, respectively) from (**b**). Bounds of boxes are the first and third quartiles for each condition. The center (median) is shown by the horizontal line in the box. Whiskers extend to 1.5 times the inter-quartile range. Outliers are shown as small circles, and minimum and maximum values are the lowest and highest circles, respectively. Read-through values and downstream FPKM were calculated as described in Methods from previously published 4sU-seq data (average of *n* = 2 biological replicates)[3]. Read-through for mock infection was defined as zero and is thus not shown. Source data are provided as a Source Data file.

leads to increased dOCRs (see next section). Second, the detection of dOCRs was likely enhanced by PAA due to technical reasons. RCs at 8 h p.i. comprise much uncompacted viral DNA, which attracts and sequesters DNA binding proteins non-specifically[33]. Thus, the transposase used for ATAC-seq likely preferentially incorporates sequencing adapters into viral DNA over cellular dOCRs as the latter is (globally) still more compacted. When PAA is added to the infected cells, the transposase is not sequestered in RCs and can better sample the loosely packaged dOCRs. Since PAA also prevents viral late gene expression, we conclude that neither viral late gene expression nor sequestration of a cellular factor to viral RCs is required for dOCR induction.

## dOCRs arise upon strong transcriptional activity downstream of genes

To assess the role of individual viral genes regarding the induction of dOCRs, we first aimed to identify a subset of cellular genes that showed

strong and consistent dOCR induction upon infection with both virus strains. To this end, we performed a hierarchical clustering analysis of dOCR lengths in mock, WT, and WT + PAA infection (Fig. 1b). The cutoff on the dendrogram was chosen such that these clusters were obtained as separate clusters, resulting in a total of nine different gene clusters. Three of these clusters (Fig. 1b, Clusters 2, 5, and 6, orange to red bars) showed dOCR induction, while six (blue and green bars in Fig. 1b) exhibited no induction of dOCRs during infection. Cluster 5 (305 genes, dark red bar in Fig. 1b) exhibited the strongest induction of dOCRs independent of the virus strain. In contrast, Clusters 2 (290 genes, orange) and 6 (701 genes, red) showed weaker dOCR induction, which was nevertheless clearly visible upon PAA treatment for both strains. Notably, these three clusters differed in the presence of dOCRs prior to infection. Here, Clusters 5 and 6 already exhibited short dOCRs prior to infection that increased in length and thus extended significantly further downstream upon infection. dOCR lengths were particularly well correlated between replicates, in particular for genes

and conditions with strong dOCR induction, i.e., genes in Clusters 2, 5, and 6 and upon WT infection with PAA treatment (both strains) and WT strain 17 infection (Supplementary Fig. 1b). For genes and conditions without dOCR induction, i.e., the remaining six clusters and mock infection, variability between replicates was much higher, likely as small absolute changes due to technical and biological noise in small values (i.e., few and short dOCRs) result in large relative changes. Correlations in dOCR lengths were also high between different conditions with dOCR induction, even though higher dOCR lengths were observed with PAA treatment compared to the corresponding untreated samples and in WT strain 17 infection compared to WT strain F (Supplementary Fig. 1c). Notwithstanding the differences in how strongly dOCRs are induced for affected genes, this confirms high overlaps in genes with or without dOCR induction between different *HSV-1* strains and ± PAA treatment.

To investigate the cause of these differences in the induction of dOCRs between the different gene clusters, we made use of our previously published 4sU-seq time-course of the first 8 h of WT strain 17 infection[3]. We analyzed (i) the percentage of read-through transcription (defined as the difference between *HSV-1* and mock infection of the following ratio: FPKM within 5 kb downstream of the gene 3′end/ gene FPKM × 100, negative values set to 0) and (ii) the absolute extent of transcriptional read-through activity (FPKM within 5 kb downstream of the gene 3′end, denoted as 'downstream FPKM'). While this showed statistically significant increases in the percentage of read-through transcription for Clusters 2 and 5 compared to all other analyzed genes (one-sided Wilcoxon rank sum test, $p < 0.0005$) at least for some timepoints, the differences were nevertheless small (fold-change of medians < 1.35). Please note that by definition, read-through in mock infection was set to zero and thus not shown. Cluster 2 (orange, weak dOCR induction) showed the highest median percentage of read-through transcription of all clusters (Fig. 1c). In contrast, downstream transcriptional activity was substantially greater in Cluster 5 (dark red, highest dOCR induction, $p < 0.0005$) than in all other clusters (Fig. 1d). This was matched by higher gene expression levels (gene FPKM) in Cluster 5 already prior to infection, which was also observed for Cluster 6 (Supplementary Fig. 1e, $p < 0.0005$). Cluster 5 thus comprises the most strongly expressed cellular genes with the highest absolute levels of downstream transcription due to DoTT. Clusters 2 and 6 also showed slightly elevated downstream transcriptional activity compared to clusters without apparent induction of dOCRs, but this was only significant for Cluster 2 (Fig. 1d, $p < 0.0005$). Accordingly, the absolute extent of transcriptional activity downstream of the respective genes rather than the percentage of read-through transcription determines the extent of dOCR induction. This also explains why genes with a high percentage of read-through transcription but relatively low gene expression (Cluster 2), as well as genes with a moderate percentage of read-through transcription but higher gene expression (Cluster 6), exhibit some induction of dOCRs. Strikingly, induction of open chromatin predominantly occurred downstream of affected gene 3′ ends, although transcription of the respective gene bodies was at least as high as downstream transcription.

As transcription downstream of genes is very low prior to infection, median downstream FPKM in mock infection is 3- (without PAA) to 13-fold (with PAA) lower than in WT infection, even in total RNA (see Supplementary Fig. 2b). Thus, analysis of total RNA, which requires <1 μg of input RNA, instead of 4sU-RNA, which requires >30 μg, is both sufficient and more economical to quantitatively assess transcriptional activity downstream of genes during infection. We thus performed total RNA-seq in parallel to ATAC-seq for mock and WT strain F infection ± PAA to confirm the results from the 4sU-seq time-course (Supplementary Fig. 2). Moreover, we found that dOCR lengths were significantly correlated to downstream transcriptional activity across all 4162 analyzed genes both in total (Fig. 2a, Supplementary Fig. 3a–c) and 4sU-RNA (Fig. 2b) in WT *HSV-1* infection, in particular upon PAA

treatment. In the absence of both read-through and dOCR induction in mock infection, no correlation was observed. While the correlation in *HSV-1* infection was not perfect, it nevertheless confirmed a general trend with longer dOCRs observed for genes with higher downstream transcriptional activity, which were enriched for Cluster 5 genes (Supplementary Fig. 3d–f). Notably, increasing the downstream window size from 5 to 10 kb for calculating downstream FPKM led to similar but slightly higher correlation coefficients (Supplementary Fig. 3g–j). To assess whether transcription extended throughout the full length of dOCRs, we visualized read coverage in total RNA-seq data for mock and WT strain F infection ± PAA in 1 kb windows throughout the dOCR regions identified for genes with strong dOCR induction, i.e., Cluster 5 genes, in 12 h p.i. WT strain F infection + PAA (Supplementary Fig. 4). This showed extensive transcription throughout the vast majority of dOCRs at both 8 h and 12 h p.i. *HSV-1* infection, in particular with PAA treatment, but not in uninfected cells (Supplementary Fig. 4). Notably, dOCRs can also extend into the gene body of downstream genes upon read-in transcription (as exemplified by read-in transcription from the *SRSF3* gene into the *CDKN1A* gene, Supplementary Fig. 1a). To exclude that dOCR arises due to increased transcriptional activity of other genes located within the respective dOCRs, we further restricted this analysis to the 103 genes in Cluster 5 with no known protein-coding gene or lincRNA within the first 50 kb downstream of their gene 3′ end (Supplementary Fig. 5). This confirmed our observations made for all Cluster 5 genes and demonstrated extensive transcription throughout dOCRs in *HSV-1* but not mock infection. We conclude that dOCRs selectively arise as a consequence of *HSV-1*-induced DoTT when strong transcriptional activity extends downstream of genes beyond affected poly(A) sites. Furthermore, inhibition of viral DNA replication by PAA enhances the induction of dOCRs due to the reduced shut-off of host transcription.

## ICP22 is required for the induction of dOCRs

As we established that PAA treatment leads to increased dOCRs in WT *HSV-1* infection, this effectively precludes a role for most viral late proteins in dOCR induction, except for a few high-copy tegument proteins. To identify the viral gene responsible for dOCR induction, we thus performed ATAC-seq for HFFF infected with a range of single gene deletion mutants ($n = 2$). This included null mutants of the immediate early genes ICP0, ICP22 (R325), and ICP27, as well as of the virion host shut-off protein (vhs), which is expressed late during infection but is delivered to infected cells by the incoming virions. Due to the attenuated progression of ΔICP0 and ΔICP22 infections, ATAC-seq was performed at 12 h p.i. for these mutants. For WT and the other mutants, infection was performed at 8 h p.i. For ΔICP22 infection, we also included PAA treatment as we hypothesized that the loss of serine 2 phosphorylation of RNA polymerase II (Ser-2P) during *HSV-1* infection might play a role in the induction of dOCRs. This loss of Ser-2P results from the combined effects of both an ICP22-dependent and an unknown ICP22-independent, viral late gene-dependent mechanism[34]. To analyze the induction of dOCRs in the single-gene deletion mutants, we first focused on genes in Cluster 5. Infection with ΔICP0 and Δvhs showed induction of dOCRs at levels comparable to WT strain 17 (Fig. 2c). Strikingly, infection with the ICP22 deletion mutant did not result in any detectable induction of dOCRs even upon PAA treatment. This was confirmed by additional ATAC-seq experiments with 8 h and 12 h p.i. ±PAA treatment both for the ΔICP22 mutant and its parental WT strain F ($n = 2$) (Fig. 2d) as well as using a ΔICP22 mutant derived from KOS1.1[35] and from BAC-derived WT strain 17 ($n = 2$, Supplementary Fig. 6a, b). This also confirmed dOCR induction by a third *HSV-1* strain, namely KOS 1.1, although slightly less prominent compared to strains F and 17 (Supplementary Fig. 6a). To check whether ΔICP22 infection still induced transcription downstream of genes, we sequenced the total RNA samples harvested in parallel to the ATAC-seq samples for mock, WT strain F and ΔICP22 infection (8 and 12 h p.i.)

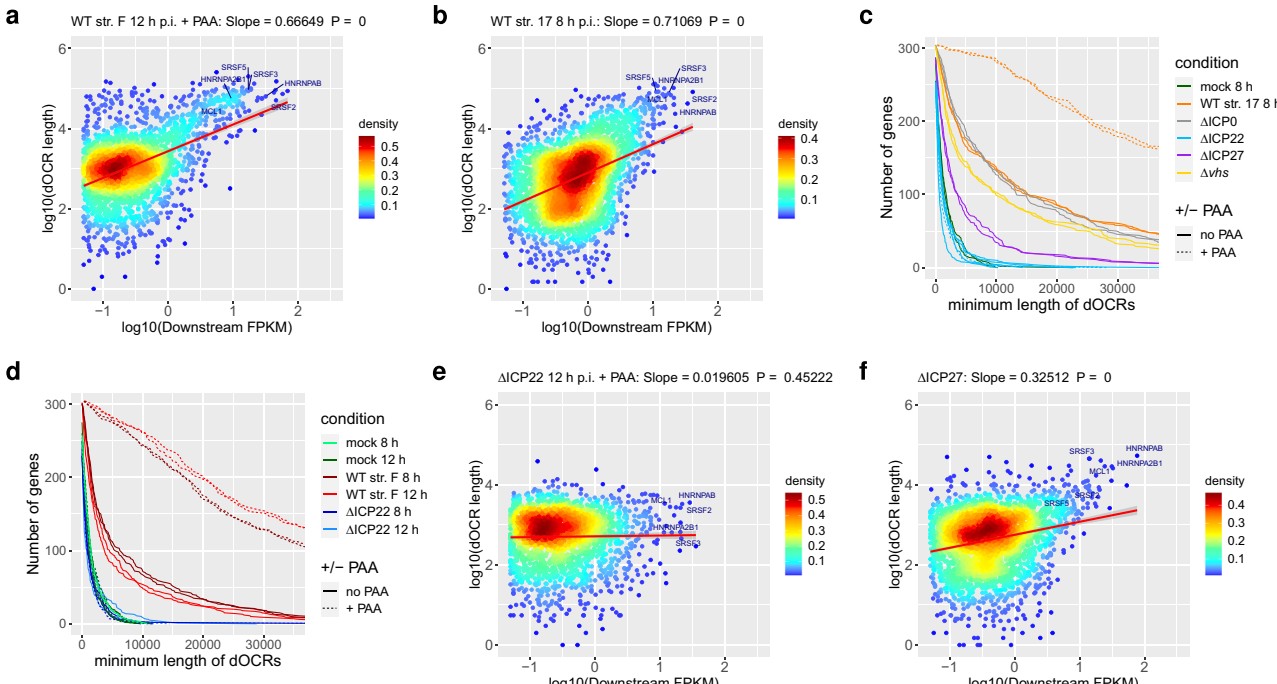

**Fig. 2 | ICP22 is required for the induction of dOCRs. a, b** Scatter plots correlating downstream FPKM against dOCR length (average of two replicates) in total RNA for WT strain F (**a**) and in 4sU-RNA for WT strain 17 (**b**) for all analyzed genes with a downstream FPKM ≥ 0.05. Colors indicate the density of points from high (red) to low (blue). The red line indicates a linear fit of log10(dOCR length) against log10(downstream FPKM). The slope of the fit and *p*-values for the slope of the linear regression estimate being ≠ 0 (two-sided test) were calculated using the *lm* function in R and are indicated on top of each figure. The error bands around the red line indicate the 95% confidence level interval for predictions from the *lm* linear model. Example genes with high induction of dOCRs in *HSV-1* infection are highlighted. **c, d** Number of genes in Cluster 5 from Fig. 1b for which dOCRs reach at least a length greater than the value indicated on the *x*-axis for mock, WT strain 17 (**c**), WT strain F (**d**), ΔICP0 (**c**), ΔICP22 (**c, d**), ΔICP27 (**c**) and Δvhs infection (**c**). To avoid having to define a threshold on whether or not a particular dOCR length for a gene is considered dOCR induction, we visualize dOCR lengths in each condition for all 305 Cluster 5 genes (excluding only those with a dOCR length = 0 in a particular condition). This depicts whether the number of genes with longer dOCRs was generally increased or not in the respective experimental condition. Results are shown separately for two biological replicates. All infections in **d** were performed with and without PAA, while in **c** PAA treatment was only performed for WT strain 17 and ΔICP22 infection (as indicated by solid (no PAA) or dashed (+PAA) lines). **e, f** Scatter plots as in (**a, b**) correlating downstream FPKM against dOCR length in total RNA for 12 h p.i. ΔICP22 infection +PAA (**e**) and in 4sU-RNA for ΔICP27 infection (**f**). Scatter plots for other analyzed conditions are shown in Supplementary Figs. 3 and 7. Source data are provided as a Source Data file.

±PAA (total RNA-seq, *n* = 2). This confirmed the presence of extensive DoTT and strong downstream transcriptional activity in ΔICP22 infection in the dOCR regions observed in WT infection (Supplementary Fig. 6c, d). Despite strong DoTT in ΔICP22 infection, no correlation was observed between downstream transcriptional activity and dOCR length across all 4162 analyzed genes irrespective of PAA treatment in either matched total RNA samples or previously published 4sU-seq data[4] from ΔICP22-infected cells (Fig. 2e, Supplementary Fig. 7a–e). In contrast, this correlation was observed in ΔICP0 and Δvhs infection for previously published 4sU-seq data (Supplementary Fig. 7f, g). We conclude that ICP22, but not ICP0 or vhs, is required for the induction of dOCRs.

Despite the key role of ICP27 in mediating *HSV-1*-induced read-through, ΔICP27 infection still induced dOCRs in Cluster 5 genes, albeit to a lesser extent (Fig. 2c). This is consistent with the residual, presumably stress-induced read-through transcription observed in ΔICP27 infection[4]. Accordingly, downstream transcriptional activity taken from previously published 4sU-seq data of ΔICP27 infection[4] correlated with dOCR lengths (based on all 4126 analyzed genes) albeit to a lesser extent than observed in WT infection (slope of linear regression estimate 0.71 in WT strain 17 vs. 0.33 in ΔICP27, Fig. 2b, f). This correlation further increased (to 0.37) when downstream FPKM was calculated for a 10 kb downstream window instead of the 5 kb window, a trend also observed for ΔICP0 and Δvhs infection (Supplementary Fig. 7h–j). In contrast, for ΔICP22 infection, increasing the downstream window generally drove the correlation closer to zero

(Supplementary Fig. 7k–p). We conclude that ICP27 is not required for the induction of dOCRs in *HSV-1* infection.

## ICP22 is sufficient for induction of dOCRs upon read-through transcription

Our analyses so far indicate that both ICP22 expression and transcription downstream of genes are necessary for the induction of dOCRs. To test whether ectopic ICP22 expression was also sufficient to induce dOCRs in the presence of downstream transcription, we generated telomerase-immortalized human foreskin fibroblasts (T-HFs) that express either ICP22 (T-HF-ICP22 cells) in isolation (T-HF-ICP22 cells) or in combination with ICP27 (T-HF-ICP22/ICP27 cells) upon doxycycline (Dox) exposure (Supplementary Fig. 8a–d). Co-induction of ICP27 served to induce strong downstream transcriptional activity in the T-HF-ICP22/ICP27 cells. As a control, we also included cells that express ICP27 in isolation upon Dox exposure (T-HF-ICP27 cells). We subsequently analyzed the induction of dOCRs by Omni-ATAC-seq and, in parallel quantified downstream transcriptional activity by total RNA-seq in the same experiment (*n* = 2). Omni-ATAC-seq represents a recent improvement in the ATAC-seq protocol published during the course of this study, which improves signal-to-background ratios and reduces the amount of contaminating mitochondrial reads[36,37]. As expected, Dox-induced ICP22 expression in the absence of ICP27-induced read-through transcription and Dox-induced ICP27 expression alone did not induce dOCRs (Fig. 3a, Supplementary Fig. 8e). In contrast, Dox-induced co-expression of ICP27 and ICP22 resulted in

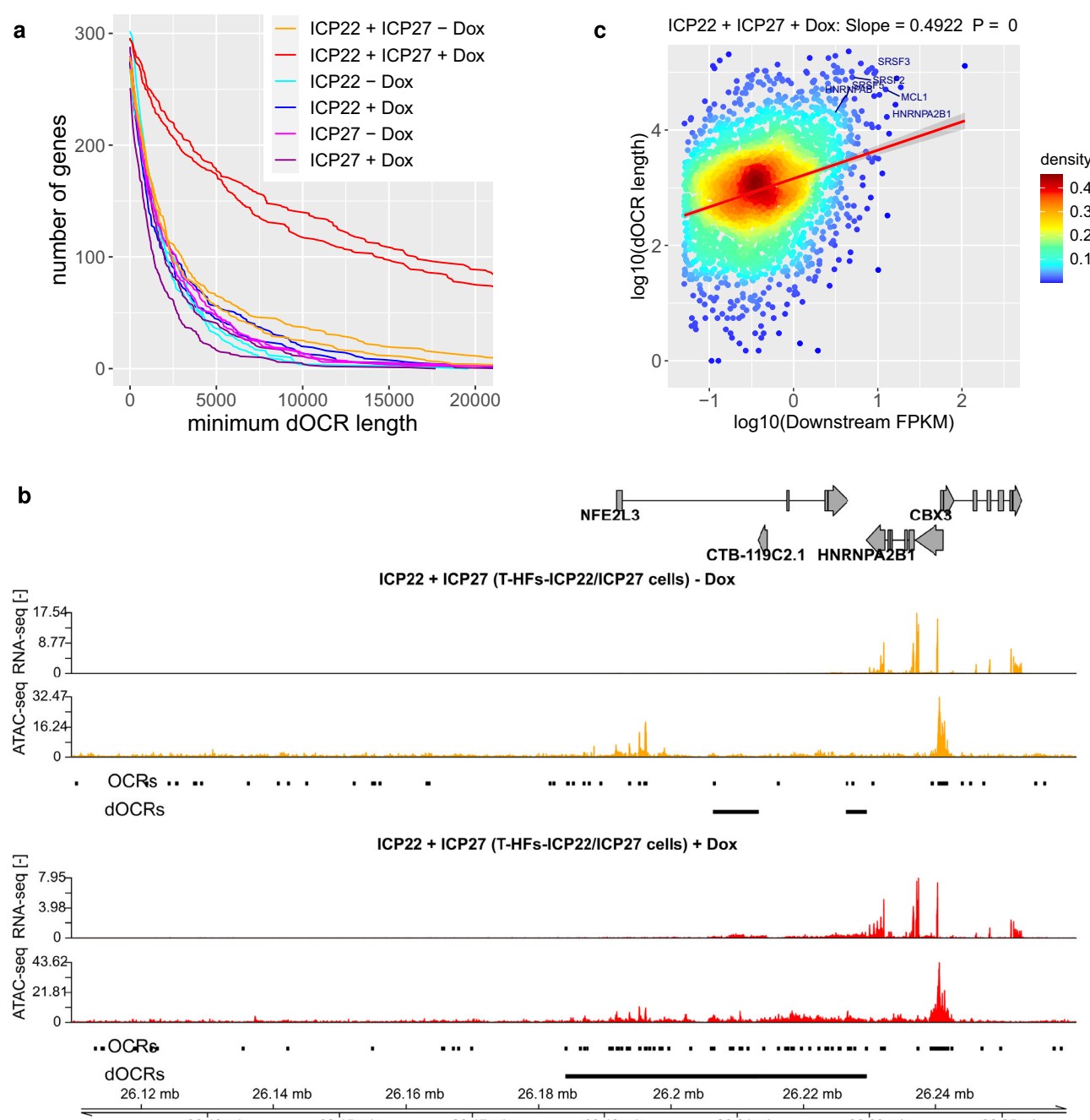

**Fig. 3 | ICP22 is sufficient to induce dOCRs upon ICP27-induced read-through transcription. a** Number of genes in Cluster 5 from Fig. 1b for which dOCRs reach at least a length greater than the value indicated on the *x*-axis in T-HF-ICP22 cells, T-HF-ICP27 cells, and T-HF-ICP22/ICP27 cells ± Dox treatment. **b** Example gene (*HNRNPA2B1*) showing induction of dOCRs after Dox-induced ICP22 and ICP27 expression in T-HF-ICP22/ICP27 cells. Tracks show total RNA-seq (strand-specific) and Omni-ATAC-seq (non-strand-specific) read coverage (normalized to a total number of mapped human reads; averaged between replicates). Below each Omni-ATAC-seq track, the figure shows open chromatin regions (OCRs) identified with F-Seq as well as the dOCR regions calculated from the OCRs as described in Methods. For simplification, OCRs and dOCRs are shown only for the first replicate. Gene annotation is indicated at the top. Boxes represent exons, lines introns, and gene direction is indicated by arrowheads. Genomic coordinates are shown at the bottom. **c** Scatter plot correlating downstream FPKM in total RNA against dOCR length (average of two replicates) for Dox-induced combined ICP22 and ICP27 expression (T-HF-ICP22/ICP27 cells + Dox). Shown are all analyzed genes with a downstream FPKM ≥ 0.05. Colors indicate the density of points from high (red) to low (blue). The red line indicates a linear fit of log10(dOCR length) against log10(downstream FPKM). The slope of the fit and *p*-values for the slope of the linear regression estimate being ≠ 0 were calculated using the *lm* function in R and are indicated on top of each figure. The error bands around the red line indicate the 95% confidence level interval for predictions from the *lm* linear model. Example genes with high induction of dOCRs in *HSV-1* infection are highlighted. The corresponding scatter plot for T-HF-ICP22/ICP27 cells without Dox treatment is shown in Supplementary Fig. 8f. Source data are provided as a Source Data file.

extensive induction of dOCRs (Fig. 3a, b, Supplementary Fig. 8e), which correlated with transcriptional activity downstream of genes (Fig. 3c). Transcription was observed across dOCRs for Cluster 5 genes (Supplementary Fig. 8g), with transcription decreasing with increasing distance from gene 3′ends. The same was observed when restricting the analysis to the 103 genes in Cluster 5 with no known protein-coding or lincRNA gene within the first 50 kb downstream of their gene 3′ end (Supplementary Fig. 8h). In the absence of Dox exposure, neither

dOCR induction nor read-through was observed in T-HF-ICP22/ICP27 cells (Fig. 3a, b, Supplementary Fig. 8e, f). Our total RNA-seq data also confirmed previous findings from HeLa cells, which demonstrated that ectopic expression of ICP27 is sufficient to disrupt transcription termination[4]. We conclude that ICP22 is sufficient for the induction of dOCRs upon ICP27-induced read-through transcription.

## Induction of dOCRs is associated with a loss of histones downstream of genes

We hypothesized that the induction of dOCRs was due to impaired histone repositioning in the wake of Pol II read-through transcription into downstream genomic regions. To test this hypothesis, we first analyzed genome-wide occupancy of histone H3 as well as the two major histone marks associated with heterochromatic regions (H3K27me3) or active transcription (H3K36me3) in uninfected and WT strain 17-infected (8 h p.i., without PAA) cells by ChIPmentation ($n = 2$ or 3; see Methods for details, Supplementary Data 2). A metagene analysis from −3 kb upstream of the TSS to 100 kb downstream of the transcription termination site (TTS) for all protein-coding genes showed the expected distributions (Supplementary Fig. 9a–c). For metagene analyses, occupancy profiles for each gene and replicate were first normalized to a sum of 1 before averaging across genes and replicates to avoid biases due to differences in gene expression (see Methods). Thus, these profiles represent relative distributions of histones across the gene and downstream regions. Notably, histone modifications were not normalized to H3 occupancy in these analyses. H3 was strongly depleted at promoters and slightly depleted on gene bodies but uniformly present in intergenic regions, while H3K27me3 was also strongly depleted on gene bodies. In the absence of any particular enrichment of H3 and H3K27me3, few peaks were identified for H3 and H3K27me3, and these were present mostly in intergenic regions (Supplementary Data 2). In contrast, H3K36me3 was strongly enriched on gene bodies but also depleted at gene promoters. To identify differences in histone and histone modification occupancy between mock and HSV-1 infection associated with dOCRs, we performed metagene analysis separately for genes with strong induction of dOCRs (Cluster 5, Fig. 4a–c) and genes without dOCR induction (= all genes except for Clusters 2, 5, and 6, Supplementary Fig. 9d–f). However, this did not reveal any significant virus-induced changes in occupancy of H3 or H3K27me3 up- or downstream of the TTS for genes in Cluster 5. For the active transcription mark H3K36me3, a reduction during HSV-1 infection was observed in the 5 kb upstream of the TTS (Fig. 4c), which was statistically significant at some positions (Wilcoxon test, $p < 0.05$). It is noteworthy, however, that this was also observed for genes without dOCR induction, although not as pronounced. In contrast, genes without dOCR induction showed a small but significant increase in H3K36me3 downstream of the TTS. While we also observed some relative enrichment around promoters for H3, H3K27me3, and H3K36me3, this was not specific to Cluster 5 genes and was likely associated with the global loss of host transcriptional activity and reduction of promoter-proximal Pol II pausing in HSV-1 infection[38,39].

To confirm the results from the metagene analyses, we performed a genome-wide differential analysis of H3, H3K27me3, and H3K36me3 in HSV-1 infection compared to mock for promoter regions (±1.5 kb around TSS), gene bodies excluding promoter regions (TSS + 1.5 kb to TTS) and downstream regions (TTS to TTS + 25 kb) for all protein-coding and lincRNA genes. Consistent with the metagene analyses, we observed small but highly significant increases in H3, H3K27me3, and H3K36me3 at gene promoters (median log2 fold-changes 0.2, 0.09, 0.06) compared to gene body and downstream regions (Wilcoxon test, $p < 10^{-10}$). However, as no significant differences in gene promoters were observed between our 9 clusters, the respective changes are thus unlikely to be linked to dOCRs. Interestingly, however, downstream regions of Cluster 5 genes, but not of other clusters, tended to show a small reduction in H3K36me3 (median log2 fold-

change −0.05) but not H3 or H3K27me3. This reduction was statistically significant compared to all protein-coding and lincRNA genes or all other genes included in our analysis (Wilcoxon test, multiple testing adjusted $p < 0.05$). In summary, we found no differences in the distribution of H3 or H3K27me3 following HSV-1 infection that could be linked to dOCR induction. However, small differences in H3K36me3 were observed downstream of genes with strong dOCR induction.

While an increase in chromatin accessibility should still be readily detectable by ATAC-seq even when only a very small percentage of cells is still transcribing the genes by 8 h p.i. (and thus induce dOCRs), a loss in histone occupancy would likely be masked by the cells not transcribing the respective genes anymore. Thus, to prevent the virus-induced sequestration of Pol II to RCs and alleviate the reduction in host transcriptional activity[33], we repeated the ChIPmentation experiments upon PAA treatment (mock and WT strain 17 infection at 8 h p.i.; both including 8 h of PAA treatment). This time, we also analyzed histones H1 and H4 in addition to H3, H3K27me3, and H3K36me3. Quality of the histone ChIPmentation data was again confirmed by metagene analyses on all protein-coding genes showing the expected occupancy profiles with a strong depletion of H1, H3, and H4 at transcription start sites (TSSs) compared to gene bodies and, in particular, downstream regions (Supplementary Fig. 10a–c). Strikingly, for genes with strong induction of dOCRs (Cluster 5), all three histones showed a reduction in coverage in WT strain 17 infection with PAA treatment compared to the uninfected cells starting around or slightly upstream of the TTS and extending for about 25 kb downstream of the TTS (Fig. 4d, e, Supplementary Fig. 11a). While this only reached significance (Wilcoxon test, $p < 0.05$) for some of these positions, it was observed consistently for all three histones. Furthermore, no such reduction was observed for genes without induction of dOCRs (Supplementary Fig. 11b–d). The increase in H1, H3, and H4 observed in WT strain 17 infection more downstream (>50 kb) is likely due to the normalization procedure for individual gene curves in the metagene analysis (see above). The genome-wide differential analysis for HSV-1 infection compared to mock showed a reduction in all three histones selectively downstream of Cluster 5 genes (median log2 fold-change −0.4 to −0.3, Wilcoxon test compared to both all protein-coding and lincRNA genes and all other clusters $p < 10^{-33}$) and to a lesser degree for H3 and H4 downstream of Cluster 6 genes (Supplementary Fig. 11g, h). Interestingly, Cluster 5 genes also tended to show a reduction within gene bodies for all three histones (median log2 fold-change −0.2 to −0.3, Supplementary Fig. 11i, j).

The histone modification marks exhibited distributions consistent with their association with heterochromatic or transcribed regions, with H3K27me3 being depleted on gene bodies (Supplementary Fig. 9e) and H3K36me3 (Supplementary Fig. 9f) being enriched. Notably, H3K27me3 showed a highly significant increase in gene bodies for all protein-coding genes and genes without dOCR induction (Supplementary Figs. 10d and 11e). While the increase was also observed for Cluster 5 genes (Fig. 4f), it was not significant. This increase in H3K27me3 is likely due to globally reduced host transcription during HSV-1 infection, even with PAA treatment. For both histone modification marks, a reduction was observed for Cluster 5 genes selectively in the region from the TTS or upstream of the TTS to around 25 kb downstream of the TTS (Fig. 4f, g). This is consistent with the similar reduction for histone H3 and was not observed for genes without induction of dOCRs (Supplementary Fig. 11e, f). Interestingly, PAA treatment even enhanced the HSV-1-induced reduction in H3K36me3 within the 5 kb upstream of the TTS selectively for Cluster 5 genes. As no such loss was observed for H3K27me3, this does not result from a global loss of H3 in this region. The underlying molecular mechanism remains unclear. Consistent with results on H3, the genome-wide differential analysis showed a reduction in both H3K36me3 and H3K27me3 downstream of Cluster 5 genes (median log2 fold-changes −0.6 and −0.2, respectively,

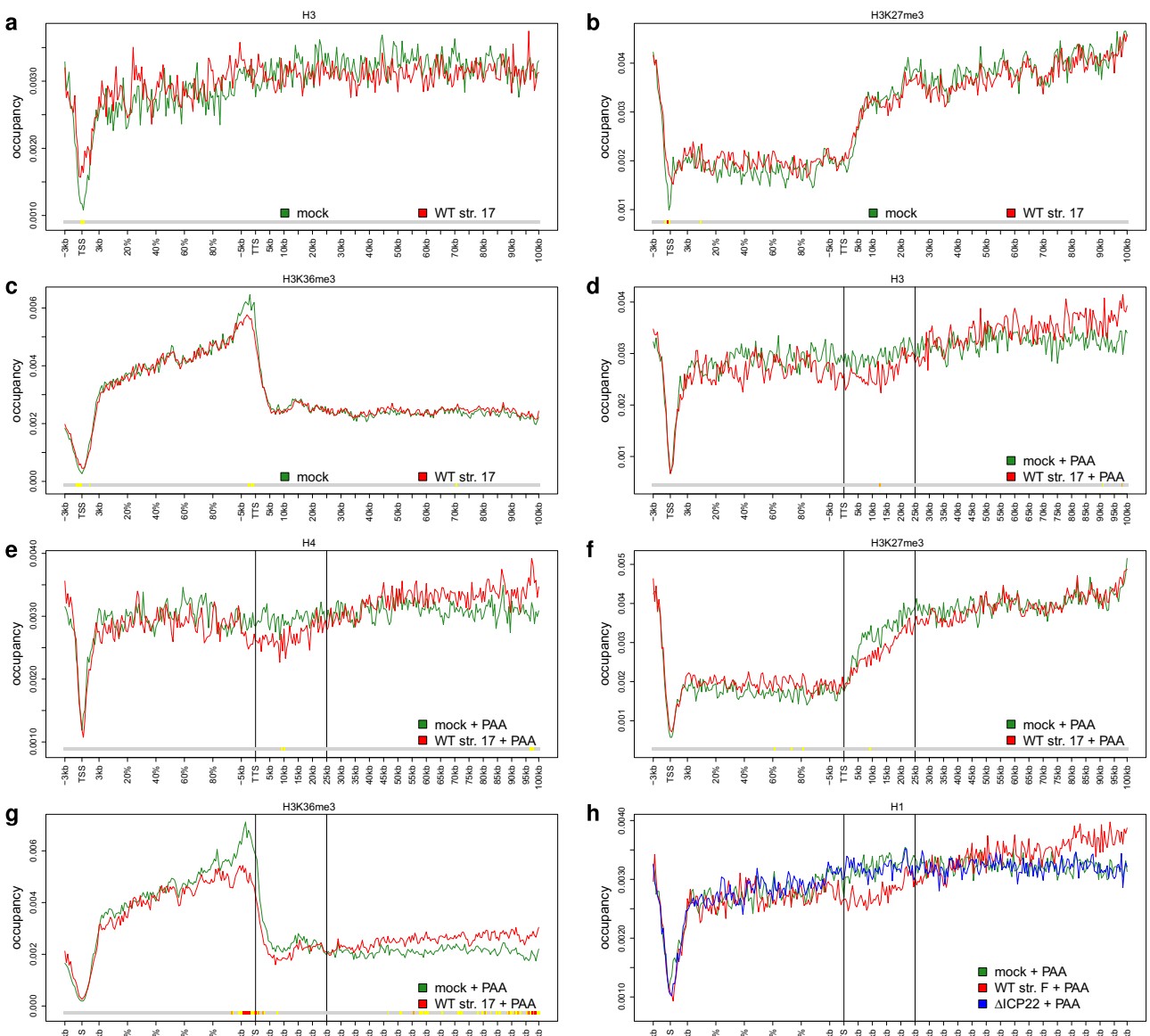

**Fig. 4 | Alterations in histone distribution associated with the induction of dOCRs. a**–**g** Metagene plots (see Methods) showing the distribution of (**a**, **d**) H3, (**e**) H4, (**b**, **f**) H3K27me3, and (**c**, **g**) H3K36me3 for genes with strong induction of dOCRs, i.e., Cluster 5 genes from Fig. 1b, in mock and WT strain 17 infections without (**a**–**c**) and with PAA treatment (**d**–**g**). Metagene plots for H1 in mock and WT strain 17 infections with PAA treatment for Cluster 5 and corresponding metagene plots for genes without induction of dOCRs are shown in Supplementary Fig. 11a–f. The color track at the bottom of each subfigure indicates the significance of paired two-sided Wilcoxon tests comparing the normalized transcript coverages of genes

for each position between mock and WT infection. *P*-values are adjusted for multiple testing with the Bonferroni method within each subfigure; color code: red = adj. *P*-value ≤ $10^{-5}$; orange = adj. *P*-value ≤ $10^{-3}$; yellow: adj. *P*-value ≤ 0.05. **h** Metagene plots showing the distribution of H1 in mock, WT strain F and ΔICP22 infection for genes with strong dOCR induction, i.e., Cluster 5 genes. *P*-values for pairwise comparisons between mock and WT strain F infection, WT strain F and ΔICP22 infection, and mock and ΔICP22 infection, respectively, were calculated as for (**a**–**g**) and are shown in Supplementary Fig. 12b–d. Source data are provided as a Source Data file.

Supplementary Fig. 11k, l). This was particularly pronounced for H3K36me3, which also showed a pronounced reduction in gene bodies (Supplementary Fig. 11m). In contrast, H3K27me3 tended to be increased in gene bodies for all clusters but least so in Cluster 5 (Supplementary Fig. 11n).

To confirm that the alterations in nucleosome abundance downstream of the TTS were specifically associated with induction of dOCRs, we performed ChIPmentation for H1 in mock, WT and ΔICP22 infection (both in strain F) at 8 h p.i. (Fig. 4h). We used H1 here as a proxy for total histone occupancy as it is bound at high frequency to nucleosomes in transcriptionally inactive areas (e.g., downstream of transcribed regions). This confirmed the observations for WT strain 17 infection with a selective reduction in H1 occupancy within the 25 kb

downstream of the TTS in WT infection compared to both mock and ΔICP22 infection. In contrast, H1 occupancy profiles for mock and ΔICP22 infection were highly similar. Again, no such effect was observed for genes without induction of dOCRs (Supplementary Fig. 12a). Statistical analysis confirmed the differences downstream of the TTS to be significant (Wilcoxon test, p < 0.05) at several positions in pairwise comparisons of WT and mock infection as well as WT and ΔICP22 infection (Supplementary Fig. 12b, c). No significant differences were observed between mock and ΔICP22 infection (Supplementary Fig. 12d). Similarly, the genome-wide differential analysis showed a decrease in H1 for downstream regions of Cluster 5 genes when comparing WT and mock infection (median log2 fold-change −0.3, Supplementary Fig. 12e) but no decrease when comparing

ΔICP22 infection and mock (Supplementary Fig. 12f). We conclude that induction of dOCRs in WT *HSV-1* infection is associated with alterations of histone occupancy downstream of affected genes, which are absent in ΔICP22 infection. This is consistent with a model in which ICP22 interferes with histone repositioning in the wake of Pol II passage.

## Depletion of FACT increases chromatin accessibility in an ICP22-dependent manner

The two histone chaperones FACT (SPT16/SSRP1) and SPT6 play a key role in the re-assembly of nucleosomes after the passing of Pol II[12]. During the course of this study, both histone chaperones were shown to be recruited into viral RCs by the viral ICP22 protein[17,27], which suggests some kind of functional modulation. Since PAA treatment increased dOCRs and ICP22/ICP27 expression in isolation were sufficient for dOCR induction, this excludes sequestration of ICP22-binding factors to RCs as a mechanism of dOCR induction. However, since functional inhibition of FACT or SPT6 on host cell chromatin by ICP22 could still explain the induction of dOCRs, we investigated whether depletion of either of the two factors would restore the induction of dOCRs upon infection with an ICP22-null mutant. For this purpose, we generated T-HF cells with Dox-inducible artificial miRNA-mediated knockdown of SSRP1 and SPT6[40]. For both cellular proteins, efficient knockdown was achieved with two different miRNAs after 3 days of Dox treatment (1 μg/ml) (Supplementary Fig. 13a, b). Knockdown of SSRP1 resulted in a concomitant loss of its interaction partner SPT16, consistent with previous reports[41]. Furthermore, knock-down of SPT6 not only significantly reduced SSRP1 but also resulted in a modest reduction of Pol II levels which further declined upon *HSV-1* infection independently of ICP22. Nevertheless, knockdown of neither of the two cellular proteins had any discernable effect on viral gene expression upon high MOI infection (Supplementary Fig. 13c, d). This implies that both proteins are not required for productive *HSV-1* infection. To assess whether the respective histone chaperons play any role in *HSV-1*-mediated induction of dOCRs, we performed Omni-ATAC-seq on WT strain F-, ΔICP22- and mock-infected cells (with PAA treatment in all cases). Omni-ATAC-seq was performed on cells from the same experiment as utilized for the Western blots in Supplementary Fig. 13a–d. Neither depletion of SPT6 nor FACT (SSRP1) resulted in significant dOCR induction in ΔICP22 or mock infection (Fig. 5a–c). While knockdown of SPT6 had no effect on the extent of dOCR induction by WT *HSV-1*, depletion of FACT significantly increased dOCR induction (Fig. 5a). Down-sampling of the ATAC-seq data to the same number of cellular reads per sample confirmed that this did not result from differences in virus replication (Fig. 5b, Supplementary Fig. 13e). Analysis of RNA-seq data obtained for the same samples as the Omni-ATAC-seq data showed that this was matched by gene expression in WT *HSV-1* infection across the whole dOCR regions for Cluster 5 genes both with and without SSRP1 depletion (Supplementary Fig. 14a–d). Furthermore, this was also observed when restricting the analysis to Cluster 5 genes with no other annotated protein-coding or lincRNA gene within 50 kb downstream of the gene 3'end (Supplementary Fig. 13f, Supplementary Fig. 14e–h). Strikingly, the knockdown of FACT also led to an increase in chromatin accessibility within gene bodies for several hundred genes (examples in Fig. 5d and Supplementary Fig. 15; 282 genes with ≥2-fold increase, magenta points in Fig. 5e). Notably, Omni-ATAC-seq coverage was observed throughout dOCR regions in WT infection with and without FACT knockdown with highest levels at gene 3' ends (Supplementary Fig. 14c, d, g, h). Thus, increased dOCR induction upon FACT depletion was not due to an increase in chromatin accessibility within other genes located in dOCR regions. When looking for cluster-specific differences, we found that genes with an increase in chromatin accessibility in gene bodies upon FACT depletion in WT infection were significantly enriched for Cluster 5 (Fisher's exact test, multiple testing adjusted *p*-value < 10⁻¹⁸) but no other clusters (*p*-value > 0.05). Interestingly, however, when analyzing

the 74 genes with a ≥2-fold decrease in chromatin accessibility upon FACT depletion in WT infection (purple points in Fig. 5e), these were also enriched for Cluster 5 (*p*-value < 10⁻⁵) but no other cluster. We hypothesize that Cluster 5 genes were preferentially affected by FACT depletion as these represent the most highly expressed genes (Supplementary Fig. 1e, Supplementary Fig. 2c) and are thus likely most dependent on FACT. However, increased chromatin accessibility within gene bodies was now also observed for other genes without DoTT and corresponding dOCRs. Interestingly, these genes often already showed a slight increase in gene body chromatin accessibility in WT infection without FACT depletion (see example in Supplementary Fig. 15c). The *HSV-1*-induced increase in chromatin accessibility is thus not fully restricted to transcribed regions downstream of genes. It will be interesting to see whether this results from disrupted (incomplete) recognition of cryptic poly(A) sites in the respective gene bodies. Knockdown of FACT in mock- and ΔICP22-infected cells also resulted in a modest increase in chromatin accessibility within gene bodies for 202 and 164 genes, respectively, and a decrease for 51 and 96 genes, respectively (Fig. 5f,g). Genes with an increase were again significantly enriched for Cluster 5 genes (*p*-value < 10⁻⁷). However, in contrast to WT infection, genes with a decrease in chromatin accessibility within gene bodies upon FACT knock-down in mock and ΔICP22 infection showed no significant enrichment in any cluster. FACT-depletion-induced chromatin accessibility increases in mock- and ΔICP22-infected cells were also less prominent than observed in WT infection. We conclude that transcription of the highly expressed Cluster 5 genes may generally be more susceptible to FACT depletion and, thus, dOCR induction. In summary, FACT knockdown not only enhanced the induction of dOCRs in WT *HSV-1* infection but also alleviated its restriction to regions downstream of genes with read-through transcription. These data directly implicate FACT in the ICP22-mediated impairment of histone repositioning in the wake of Pol II.

## Discussion

Both *HSV-1* infection and various stressors trigger extensive transcription downstream of genes, but dOCRs only arise in *HSV-1* infection. Our study explains this difference by demonstrating that the viral ICP22 protein is required for the induction of dOCRs in *HSV-1* infection. Importantly, dOCR formation critically depended on transcriptional activity downstream of genes as dOCRs were only induced in *HSV-1* infection for genes with strong downstream transcription, while ICP22 expression alone, which does not induce read-through, did not induce dOCRs. The absolute level of transcriptional activity downstream of a given gene is determined by the transcriptional activity of the gene itself and the extent of failure in terminating transcription at the gene's 3'end, i.e., the extent of read-through. This explains why genes in Cluster 5 and, to a lesser extent, genes in Cluster 2 and 6 but not any of the other clusters identified in our analysis showed prominent dOCR induction upon *HSV-1* infection. For the six clusters without dOCR induction, absolute levels of transcriptional activity downstream of genes likely did not reach sufficiently high levels to induce dOCRs since they were neither particularly highly expressed nor exhibited particularly high read-through. Low transcriptional activity downstream of the respective genes during the last few hours prior to ATAC-seq analysis in a large percentage of cells thus probably explains the absence of dOCRs.

PAA treatment substantially increased the number of ATAC-seq reads mapping to the cellular genome by reducing the contribution of viral reads. However, it also increased the extent of dOCR induction due to PAA-mediated preservation of Pol II on host chromatin and, consequently, host transcriptional activity within and downstream of genes. Thus, PAA treatment is ideal for comparative analysis of dOCR induction between different virus mutants or strains, as it also prevents additional secondary effects on the progression of productive infection.

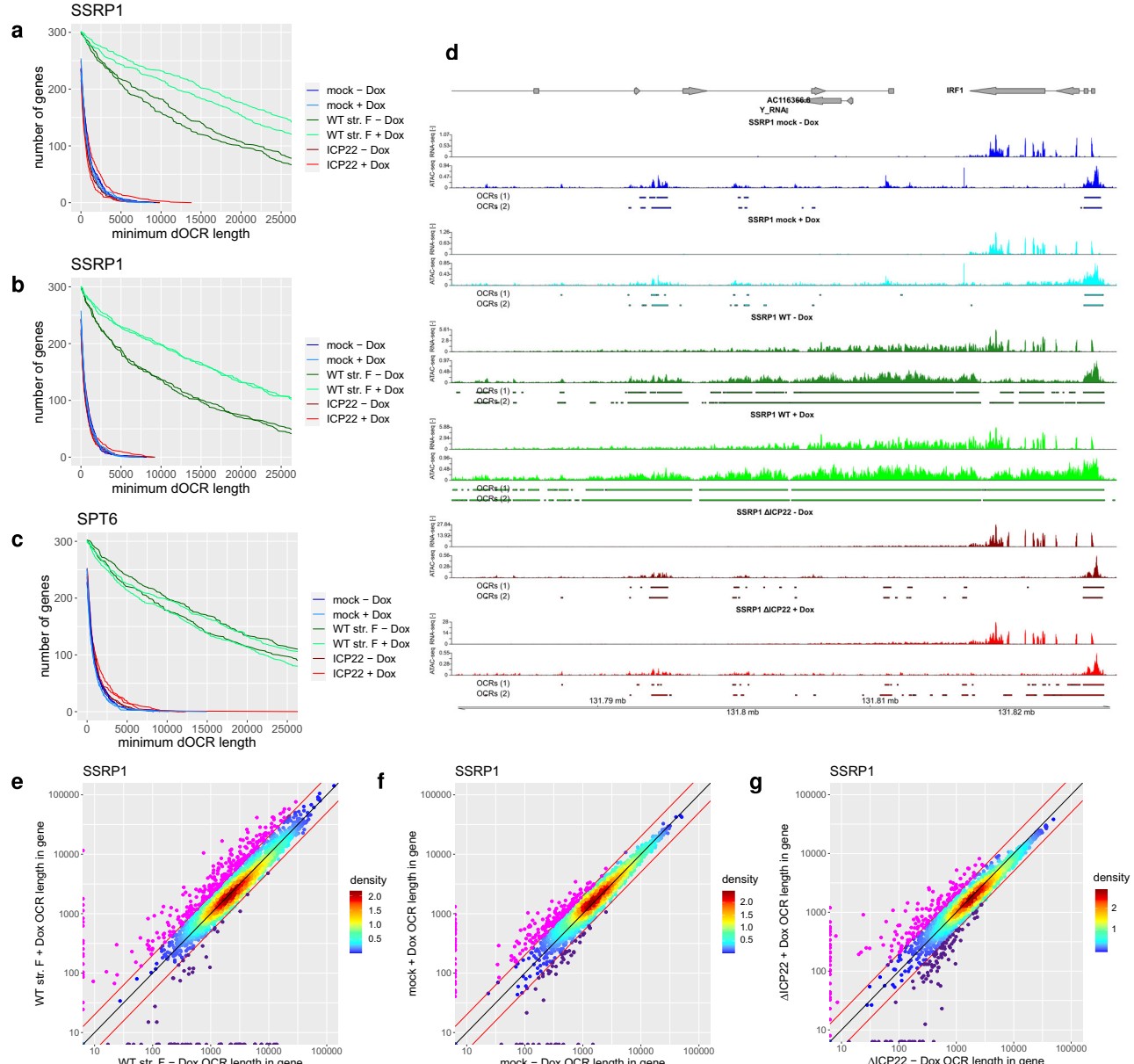

**Fig. 5 | Depletion of FACT increases chromatin accessibility in an ICP22-dependent manner. a–c** Number of genes in Cluster 5 from Fig. 1b that exhibit dOCRs with at least a length greater than the value indicated on the *x*-axis in mock, WT strain F and ΔICP22 infection with or without Dox-induced knockdown of SSRP1 **a** without and **b** with down-sampling of reads or **c** knockdown of SPT6 without down-sampling. Results for SPT6 with down-sampling of reads are shown in Supplementary Fig. 13e. **d** Example gene (*IRF1*) showing increased chromatin accessibility within the gene body in SSRP1-depleted cells in *HSV-1* infection. Tracks show total RNA-seq (strand-specific) and ATAC-seq (non-strand-specific) read coverage (normalized to a total number of mapped human reads; averaged between replicates) in mock, WT and ΔICP22 infection without and with Dox-induced SSRP1 depletion. Identified OCRs for both replicates are shown separately below the read coverage tracks. Gene annotation is indicated at the top. Boxes represent exons and lines introns, and gene direction is indicated by arrowheads. Genomic coordinates are shown at the bottom. **e–g** Scatter plots comparing the total length of open chromatin regions (OCRs) within gene bodies with and without Dox-induced knock-down of SSRP1 in **e** WT, **f** mock, and **g** ΔICP22 infection for all 4162 analyzed genes without read-in transcription. Colors indicate the density of points from high (red) to low (blue). Genes with ≥2-fold increased and reduced OCR lengths within the gene body are marked in magenta and violet, respectively. Red lines indicate a 2-fold change, and the black line depicts the diagonal. Source data are provided as a Source Data file.

The association between dOCRs and downstream transcriptional activity also explains why some of the clusters without dOCR induction, e.g., Clusters 1 and 3, showed a reduction in the already relatively short dOCR length during *HSV-1* infection, which increased again upon PAA treatment. As Pol II transcription extends beyond the poly(A) site before termination, some open chromatin can also be observed prior to infection downstream but in the proximity of the poly(A) site, in particular for highly expressed genes. Due to the virus-induced loss of host transcriptional activity[38], this downstream transcription is reduced for genes without read-through transcription, leading to a further reduction of these short dOCRs. PAA treatment prevents the extensive loss of host transcriptional activity and thus restores these short dOCRs. We would like to stress that the nine identified clusters are not set in stone but rather serve as a tool to identify the viral genes that may or may not be involved in dOCR formation. While the susceptibility of a gene's poly(A) site to disruption of transcription termination by *HSV-1* is likely an inherent feature, dOCR induction also depends on the expression level of the respective gene in the cells under study. This will vary between different cell types and conditions.

We ruled out that other viral factors are required for dOCR induction by analyzing chromatin structure in PAA-treated infected cells, which reduces or abolishes the expression of late genes, and by using several *HSV-1* null mutants. Both the selective analysis of genes with strong dOCR induction in human fibroblasts, i.e., Cluster 5, as well as a global regression analysis across all 4162 cellular genes analyzed in our study, revealed that ICP22 but not ICP0, ICP27, or vhs are required for dOCR induction. Furthermore, Dox-induced expression of ICP22 in combination with ICP27 confirmed that ICP22 is sufficient for dOCR induction upon disruption of transcription termination by ICP27. Notably, ICP27 alone did not induce dOCRs despite inducing transcription downstream of genes. Finally, ICP22 alone did not induce dOCRs in the absence of ICP27-mediated read-through transcription. Interestingly, infection with an ICP27-null mutant still induced a reduced but nevertheless significant induction of dOCRs. This is consistent with the reduced but nevertheless detectable levels of transcription downstream of genes in infection with an ICP27-null mutant[3], which likely represents a stress response. This suggests that cellular stress responses, and thus DoG transcription[7,42], may be sufficient to induce dOCRs in the presence of ICP22. It will be interesting to see whether ICP22 also triggers dOCRs in other models of impaired transcription termination.

All three strains included in our analysis (strains 17, F and KOS) induced dOCRs, although to different extents. dOCR induction was most pronounced for strain 17, followed by strain F and KOS 1.1 (Supplementary Fig. 6a). There is very little strain variation in the ICP22 sequence across *HSV-1* strains with KOS1.1 (KT887224.1) showing only six amino acid substitutions to the identical strain F (GU734771.1) and strain 17 (NC_001806.2) ICP22 sequences. The ICP22 mutants, which we employed in this study, either lack the ICP22 protein completely (KOS1.1 and strain 17) or just its C-terminal 220 aa (strain F, R325). This implies that the N-terminus of ICP22, which is important for VICE domain formation[43], is not sufficient for the induction of dOCRs. The N-terminus of ICP22 harbors five of the six mutations in KOS1.1. The respective mutations may nevertheless contribute to the reduced dOCR induction of this strain compared to strain F and 17. The dependency of dOCR induction on strong downstream transcriptional activity observed in our data supports a model in which the extent of dOCR induction by different strains is determined by the absolute extent of downstream transcriptional activity. The latter is defined by the extent of ICP27-induced disruption of transcription termination, i.e., the percentage of transcripts showing read-through and the overall extent of virus-induced transcriptional shut-off. In summary, the observed strain-specific differences in the extent of dOCRs likely reflect differences in the extent of downstream transcription. These differences result from the cumulative effects of sequence variations between the different strains in viral proteins other than ICP22 and ICP27.

The C-terminal region of ICP22 deleted in strain F was required for dOCR induction and includes the core sequence (motif 1) present in all α-herpesvirus U$_S$1 homologs, which harbors the CDK9-binding site and secondary structure elements important for folding and functioning of this region[44]. However, we cannot exclude that the deletion alters the conformation, localization, or association of the ICP22 N-terminus with other cellular factors. Mutational work and interaction studies on ICP22 are ongoing to identify the underlying molecular mechanism. Since ectopic expression of ICP22 was sufficient to induce strong dOCRs upon co-expression with ICP27, phosphorylation of ICP22 by the viral pU$_L$13 or pU$_S$3 kinases is not required for the induction of dOCRs. Even in the absence of pU$_L$13 and pU$_S$3, ectopically expressed ICP22 is still extensively modified by cellular factors[23]. Post-translational modifications of ICP22 may thus still be important for dOCR induction. Similarly, we can rule out that ICP22-mediated recruitment of cellular factors into viral RCs plays a role as a mechanism to induce dOCRs by inhibiting their function on the cellular chromatin.

ChIPmentation for major histones and common histone marks revealed a selective decrease in histone occupancy within the first ≈25 kb downstream of genes with dOCR induction. This is consistent with the increased chromatin accessibility observed in ATAC-seq experiments and was dependent on the presence of ICP22. The concordant loss of total H1, H3, and H4, as well as of both the activating and inhibitory histone marks H3K36me3 and H3K27me, respectively, indicates that impaired histone repositioning is not restricted to histones with specific histone marks but affects histone repositioning in general. The SSRP1–SPT16 heterodimer (FACT) is well described to bind both H2A–H2B dimers[13] and H3–H4 tetramers[45] to unwrap the nucleosomal DNA. However, SSRP1 was recently shown to also bind to the linker histone H1 as a homodimer and to mediate eviction of H1, suggesting an SPT16-independent function of SSRP1[46]. How ICP22 induces dOCRs by interacting with SSRP1 and SPT16 thus remains unclear. Nevertheless, our results support a model in which ICP22 impairs histone repositioning in the wake of Pol II downstream of genes. It is important to note that the loss of histone occupancy downstream of genes with dOCR induction was only detectable when viral genome replication was inhibited by PAA. Our findings thus stress the importance of studying the function of viral immediate early gene products on the host transcriptional machinery in the absence of the *HSV-1*-induced shut-down of host transcription.

During Pol II transcription, nucleosomes are first destabilized and momentarily removed to be reassembled once again at the same position in the wake of Pol II (reviewed in[12]). This is facilitated by Pol II-associated histone chaperones, including SPT6 and FACT. During the course of this study, ICP22 was found to directly interact with FACT and recruit both FACT and SPT6 to the viral RCs[17]. However, induction of dOCRs upon ectopic expression of both ICP22 and ICP27, as well as our findings involving PAA treatment, imply that deprivation of cellular factors by sequestration to large viral RCs is not responsible for dOCR induction. Interestingly, the knockdown of FACT but not SPT6 significantly increased dOCR induction in WT *HSV-1* infection. The requirement for FACT, but not for SPT6, in nucleosome reassembly, may be explained by a model in which SPT6-mediated histone chaperoning can be compensated by FACT, which can chaperone all four core histones onto DNA. In contrast, loss of FACT activity cannot be compensated by SPT6, which only chaperones histones H3 and H4[12]. Importantly, FACT knockdown also resulted in a small but nevertheless significant increase in chromatin accessibility within gene bodies of a few hundred genes in uninfected and ΔICP22-infected cells. This was substantially increased upon WT infection and often reached similar levels as observed downstream of the respective genes. Of note, a slight increase in chromatin accessibility within gene bodies was also observable for many of these genes in WT *HSV-1* infection without FACT depletion, thereby demonstrating that the increased chromatin accessibility observed in *HSV-1* infection is not completely restricted to genomic regions downstream of genes. Interestingly, affected genes, as well as genes with a reduction of chromatin accessibility upon FACT depletion in WT infection, were significantly enriched for Cluster 5 genes, indicating that these highly expressed genes are most dependent on FACT. Our findings indicate that increased chromatin accessibility both within and downstream of genes results from a common mechanism, namely functional impairment of FACT by the combined effects of FACT knockdown and the viral ICP22 protein. More direct proof of the involvement of FACT in increased chromatin accessibility would come from FACT chromatin immunoprecipitation (ChIP-seq or ChIPmentation) experiments. However, we were unable to obtain data of sufficient quality for FACT despite multiple attempts. We nevertheless would like to propose a model in which the viral ICP22 protein, via its interaction with FACT, interferes with the function of FACT to reposition histones in the wake of Pol II. The FACT complex is thought to associate indirectly with Pol II by either HP1 or the PAF1 complex bridging the interaction (reviewed

in[12,47,48]). Furthermore, chromatin remodelers can interact with histones and FACT and likely promote FACT association with chromatin. It is thus probably not surprising that the resistance of FACT to viral inhibition by ICP22 changes during transcription and drops downstream of genes where allosteric changes in Pol II result in a loss of transcription elongation factors from Pol II before transcription termination. The selective induction of dOCRs in *HSV-1* infection thereby supports the allosteric model of transcription termination, which proposes allosteric changes in Pol II composition at the end of genes[49]. In the presence of both efficient FACT knockdown and impairment of FACT function in Pol II transcription by the viral ICP22 protein (possibly by interfering with the recruitment of FACT to Pol II), these differences are alleviated, often resulting in concordant increases in chromatin accessibility both within and downstream of genes for genes with strong dOCR induction.

Multiple different functions have been attributed to ICP22. One of the most striking is the loss of serine 2 phosphorylation (Ser-2P) of the Pol II CTD, which governs the recruitment of other cellular proteins to Pol II. We cannot exclude that viral interference with Ser-2P and resulting changes in Pol II composition, in addition to the direct interaction of ICP22 with FACT, contribute to the observed effects. Finally, a recent study identified many other cellular factors involved in transcription elongation, including P-TEFb and additional CTD kinases that interact with ICP22[27]. Manipulation of other cellular factors by ICP22 may thus contribute to dOCR induction by ICP22.

At present, we can only speculate about the functional importance of this viral interference with histone repositioning. Interestingly, efficient knockdown of neither SPT6 nor FACT did significantly impair viral protein expression during productive infection, arguing against an important role of the two factors in viral transcription. Importantly, both FACT and SPT6 represent important transcription elongation factors. Viral interference with their activities as well as their recruitment to viral RCs, is thus likely to contribute to the selective virus-induced shut-off of host transcription by interfering with transcription elongation[17]. Due to the multiple functions of ICP22, the relative contribution of FACT manipulation will be difficult to decipher. It is important to note that FACT has also been shown to play an important role during the early steps of transcription. In Drosophila, it alleviates transcription inhibition by DSIF (DRB sensitivity-inducing factor) and NELF (negative elongation factor)[50]. The ICP22-mediated induction of dOCRs may thus only represent a bystander effect of more important viral interference with key mechanisms of upstream transcription elongation. However, it may also help to increase the pool of free histones[51,52] to aid the chromatinization of incoming viral genomes at early times of infection when both ICP22 and ICP27 are expressed. In summary, our findings highlight *HSV-1* as an exciting model to study fundamental aspects of the transcriptional machinery in human cells.

## Methods

### Cell culture, treatments, and infections

Human Fetal Foreskin Fibroblasts (HFFF, purchased from ECACC), Telomerase-Immortalized Human Foreskin Fibroblasts (T-HF)[53], Baby Hamster Kidney cell line (BHK, obtained from Dr. Colin Crump, Cambridge), Human Bone Osteosarcoma Epithelial Cells (U2OS, kindly provided by Stacey Efstathiou), ICP27-complementing Vero 2-2 cell line (kindly provided by Prof. Dr. Beate Sodeik) and Human embryonic kidney 293 T cells (HEK-293T, obtained from ATCC) were cultured in Dulbecco's Modified Eagle Medium (DMEM, ThermoFisher #41966052) supplemented with 10% (v/v) Fetal Bovine Serum (FBS, Biochrom #S0115), 1× MEM Non-Essential Amino Acids (ThermoFisher #11140050) and 1% penicillin/streptomycin. All cells were incubated at 37 °C in a 5% (v/v) $CO_2$-enriched incubator. HFFFs were utilized from passages 11 to 17 for all high-throughput experiments.

This study was performed using wild-type (WT) *HSV-1* strain 17, BAC-derived *HSV-1* strain 17+Lox (kindly provided by Beate Sodeik)[54],

wild-type *HSV-1* strain F, and mutant viruses R325 (ΔICP22 C-terminal 220 amino acids, strain F[55]), wild-type KOS1.1 (kindly provided by Steven Rice)[56], vhs-inactivated mutant (Δvhs, strain 17[57]), ICP27-null mutant (ΔICP27, strain KOS[58]) and ICP0-null mutant (ΔICP0, strain 17[59]). Virus stocks were produced in BHK cells as described, except for the viruses mentioned below. Stocks of the ICP27-null mutant were produced on complementing Vero 2-2 cells[60] and ICP0-null mutant in U2OS cells. All produced viruses were Ficoll-gradient purified. For all experiments, media were collected immediately prior to inoculation (conditioned media). Cells were infected for 1 h using a multiplicity of infection (MOI) of 10 in fresh media 24 h after the last split. Subsequently, the inoculum was removed, and warm, conditioned media was applied back to the cells. The time at which inoculum was replaced with growth media was marked as the 0 h time point. To block viral DNA replication, phosphonoacetic acid (PAA, 350 μg/ml) was added in conditioned media to cultured cells after the inoculum was removed. The number of biological replicates that were performed for each experiment is indicated in the results section.

### Cell line manipulation and generation

Artificial miRNAs (amiRNAs) against SSRP1 and SPT6 were selected as described[40], cloned into a doxycycline-inducible lentiviral vector (see below), and utilized to generate T-HF cells that enable efficient, dox-inducible knock-down of the respective host proteins. Primer sequences are listed in Supplementary Data 3. Transduced T-HF cells were maintained in 5 μg/ml Puromycin. Knock-down was induced by 1 μg/ml doxycycline for 72 h with fresh Dox added at 48 h after seeding.

HA-ICP22 and HA-ICP22 + V5-ICP27 cells were generated as follows. Lentiviral vectors encoding N-terminally 3xFLAG and V5-tagged (tandem tag) *UL54* ORF under control of the doxycycline-inducible pTRE-Tight promoter were produced by cloning the corresponding ORF from the *HSV-1* genome (strain 17) via intermediate vectors into pW-TH3. The pW-TH3 vector was derived from pCW57.1 by sequential insertion of a synthetic multi-cloning site (prW64/65) and three stop codons (prW110/111) between the NheI and AgeI restriction sites. pCW57.1 was a gift from David Root (Addgene plasmid #41393; http://n2t.net/addgene:41393). pW-TH7 (3xFlag-V5-NT1) was created by amplifying the N-terminal part of NT1 from the V5-NT1 vector[61] by using primers prW196/197 and inserted back between the BamHI and EcoRI of the same V5-NT1 vector. The 3xFlag-V5-NT1 ORF was excised with EcoRI and XbaI and inserted between the EcoRI and NheI sites of pW-TH3 (now designated pW-TH9). The *UL54* ORF was amplified from the *HSV-1* genome by PCR using primers prW365/366. The PCR product was digested with BamHI and BglII and inserted into BamHI cut pW-TH9 (now designated pW-TH57). To generate the doxycycline-inducible vector with HA-tagged $U_S1$ ORF (designated as LDJ5), the vector YC1 was used as the backbone. YC1 was generated to carry the blasticidin resistance gene instead of puromycin by restriction digestion of the pW-TH3 vector with XbaI and AgeI and insertion of an hPGK.blast construct (purchased from GeneArt) via infusion cloning. The $U_S1$ ORF was amplified from the *HSV-1* genome by PCR using primers prW1656/1657, and the extracted band was cloned via infusion cloning into YC1, linearized by restriction digest with MluI and NheI. Primer sequences used for generating HA-ICP22 and V5-ICP27 cell lines are listed in Supplementary Data 3.

To generate V5-ICP27 and HA-ICP22 doxycycline-inducible cell lines, HEK-293T cells were transfected with pW-TH57 and LDJ5, respectively. Transduced T-HF cells were kept in selection with 5 μg/ml puromycin and 5 μg/ml blasticidin, respectively. To generate a V5-ICP27 + HA-ICP22 doxycycline-inducible cell line, V5-ICP27 cells were lentivirally transduced with LDJ5. Transduced T-HF cells were kept in selection with both 5 μg/ml puromycin and 5 μg/ml blasticidin. Expression of proteins was induced by 5 μg/ml doxycycline for 48 h.

## Western blots

Samples were harvested at the indicated time points by removal of growth media, followed by 1× wash with phosphate-buffered saline (PBS, Sigma-Aldrich #D8537) and lysis in 1× Laemmli buffer containing 5% (v/v) β-mercaptoethanol. Samples were sonicated and heated for 5 min at 95 °C before loading onto a Novex WedgeWell 4–20% Tris-Glycine Gel (ThermoFisher #XP04200BOX). Proteins were transferred to 0.2 μm nitrocellulose membranes (Sigma-Aldrich #GE10600001), blocked for 1 h at room temperature (RT) in 1x PBS with 0.2% Tween (PBS-T) containing 5% (w/v) milk (Carl Roth #T145.3), and probed using anti-V5 (Cell Signaling #13202, 1:1000), anti-HA clone 11 (Biolegend #16B12, 1:1000), anti-FLAG (Sigma-Aldrich # F3165, 1:1000) anti-α-Tubulin (Cell Signaling #2144, 1:1000), anti-β-Actin clone C-4 (Santa Cruz Biotechnology #sc-47778, 1:1000), anti-GAPDH (Cell Signaling #2118,1:1000), anti-SPT6 (Novus Biologicals #NB100-2582, 1:500), anti-SSRP1 (Biolegend #609710, 1:350), anti-Spt16 clone 8D2 (BioLegend #607008, 1:1000), anti-RNA Pol II 1F4B6 (Active Motif #2687513, lot 17316002, 1:1000), anti-ICP8 clone 11E2 (Santa Cruz Biotechnology #sc-53330, 1:1000), anti-gD clone DL6 (Santa Cruz Biotechnology #sc-21719, 1:1000) overnight at 4 °C at the indicated dilution. Before the addition of each antibody, blots were washed with 3× PBS-T. After incubation with either anti-rabbit–horseradish peroxidase (HRP, Sigma-Aldrich #A0545, 1:10,000), anti-mouse–HRP (Sigma-Aldrich #A9044, 1:10,000), anti-rat-HRP (Sigma-Aldrich #A5794, 1:10,000), or IRDye 680RD goat-anti-rabbit IgG (Licor #926-68071, 1:5000) and IRDye 800CW donkey anti-mouse IgG secondary antibody (Licor #926–32212, 1:5000), bands were visualized using the LI-COR Odyssey FC Imaging System. PAGERuler Plus was used as a ladder. Source images are supplied without and with a ladder and as 'ladder only' scans.

## Immunofluorescence analyses

Totally, $10^5$ HA-ICP22 and HA-ICP22 + V5-ICP27 cells were plated in 12 well-dishes with the addition of 5 μg/mL of doxycycline (Merck #AMBH2D6FB132). At 48 h post-induction, cells were fixed with 4% formaldehyde (PFA) in PBS for 15 min at RT, washed three times in PBS, and either stored at 4 °C overnight in PBS or processed immediately as follows. Cells were permeabilized in 0.5% Triton X-100 in PBS for 5–10 min and blocked in a blocking buffer (10% FBS, 0.25 M glycine, 1× PBS) for 1 h at RT. Anti-HA antibody clone F-7 (Santa Cruz Biotechnology #sc-7392, 1:1000) or anti-V5 antibody (Cell Signaling #13202, 1:500) were incubated in 10% FBS and 1× PBS for 1 h at RT. Control imaging was performed in parallel for each ATAC-seq or ChIP experiment. Briefly, cells were seeded at the same density as for the specific assay and infected the next day with an MOI of 10 for 1 h. Cells were fixed at 6 hpi with 4% PFA and processed as described. Anti-ICP4 antibody (clone 10F1, Santa Cruz Biotechnology, #sc-56986, 1:1000) was incubated for 1 h to detect ICP4. For all assays, the secondary anti-mouse IgG, Alexa Fluor 488 (ThermoFisher #A11017, 1:1000) or anti-rabbit IgG, Alexa Flour 568 (Abcam #ab175471, 1:1000), were incubated in 10% FBS in 1× PBS for 1 h at RT with 0.5 μg/mL 4′,6-diamidino-2-phenylindole (DAPI). All steps were followed by three 5 min washes in 1× PBS, after which the images were taken on a Leica DMi8 fluorescence microscope. Images were exported as tif files with 10 μm scale bars on black background.

## ATAC-seq and Omni-ATAC-seq

ATAC-seq was performed according to the original protocol starting with $1 × 10^5$ cells per condition[62]. An improved ATAC-seq protocol, Omni-ATAC-seq, was performed according to the original protocol starting with $1 × 10^5$ cells per condition[36]. For each experiment, biological duplicates were carried out. Sequencing libraries for ATAC-seq samples were prepared as specified using the Nextera DNA Library prep kit (Illumina #15028212) or using NEBNext Ultra II master mix in combination with primers made by IDT based on Illumina primers with

unique dual (UD) index adapters for both i_5 and i_7. Sequencing libraries for Omni-ATAC samples were prepared as 50 μL reactions containing: 12,5 μL DNA, 6,25 μL i_5_x and i_7_x (10 μM), and 25 μL 2× NEBNext Ultra II Q5 Master Mix (NEB #M0544). The number of cycles necessary for the library amplification was determined from the pre-amplification of transposed fragments using quantitative PCR (SYBR green). Both ATAC-seq and Omni-ATAC-seq libraries were quantified by Agilent Bioanalyzer for fragments between 150 and 1000 base pairs (bs) to quantify the level of contamination with large DNA fragments. Libraries were then pooled to the same final concentration (range 150–1000 bp), loaded onto a 1% pre-cast agarose gel, excised in the specified range, and sequenced by NextSeq 500 (Ilumina) at the Core Unit Systemmedizin, Würzburg, Germany (35 bp paired-end reads). All samples were sequenced at equimolar ratios.

## RNA-seq controls

To confirm the presence of read-through transcription, on the day of the ATAC/Omni-ATAC-seq experiment, total RNA was collected. Biological duplicates were carried out. For total RNA, cells were collected in 500 μl TRI reagent, and total RNA was isolated with Directzol-RNA Microprep Kit (Zymo Research #R1050) according to manufacturer's instructions. The following steps were performed by the Core Unit Systemmedizin, Würzburg, Germany. For the total RNA libraries, both cytoplasmic and mitochondrial rRNA species were depleted. No rRNA depletion was performed for 4sU-RNA samples as rRNA only contributes about 40–50% of reads in 4sU-RNA samples. Library preparation for sequencing was performed using the stranded TruSeq RNA-seq protocol (Illumina, San Diego, USA). Libraries were sequenced on NextSeq 500 (Ilumina). 4sU-seq data for Fig. 1c, d and Supplementary Fig. 1d were taken from[3,5] and 4sU-seq data for Fig. 2e, f and Supplementary Fig. 7g–j from[4].

## ChIPmentation, library preparation, and sequencing

Two days prior to infection, two million HFFF cells were seeded in 15 cm dishes. On the day of infection, cells had expanded to ~80% confluency. Cells were infected with the respective viruses as described in the results section ($n = 2$ for all conditions except for H3K36me3 in WT strain 17 without PAA ($n = 3$)). PAA (350 μg/mL) was added to the conditioned cell culture media that was supplied to the cells after the removal of the virus inoculum. At 8 p.i., cells were fixed by adding ChIP Cross-link Gold according to the manufacturer's instructions (Diagenode #C01019027) and subsequently with 1% PBS-buffered formaldehyde. Cells were scraped in 1 mL of ice-cold 1× PBS containing protease inhibitor cocktail (1×) (Roche #11836153001) with an additional 1 mM phenylmethylsulfonyl fluoride (PMSF). Cells were pelleted at 500g for 20 min at 4 °C. Supernatant was aspirated, and cell pellets were frozen in liquid $N_2$.

Cell pellets were resuspended in 1.5 mL 0.25 [w/v] SDS sonication buffer (10 mM Tris pH = 8.0, 0.25% [w/v] SDS, 2 mM EDTA) with 1× protease inhibitors and 1 mM additional PMSF and incubated on ice for 10 min. Cells were sonicated in fifteen 1 min intervals, 25% amplitude, with Branson Ultrasonics SonifierTM S-450 until most fragments were in the range of 200–700 bp as determined by agarose gel electrophoresis. Two million cells used for the preparation of the ChIPmentation libraries were diluted 1:1.5 with equilibration buffer (10 mM Tris, 233 mM NaCl, 1.66% [v/v] Triton X-100, 0.166% [w/v] sodium deoxycholate, 1 mM EDTA, protease inhibitors) and spun at 14,000×g for 10 min at 4 °C to pellet insoluble material. The supernatant was transferred to a new 1.5 mL screw-cap tube and topped up with RIPA-LS (10 mM Tris-HCl pH 8.0, 140 mM NaCl, 1 mM EDTA pH 8.0, 0.1% [w/v] SDS, 0.1% [w/v] sodium deoxycholate, 1% [v/v] Triton X-100, protease inhibitors) to 200 μL. Input and gel samples were preserved. Lysates were incubated with 1:100/IP of anti-H1 antibody (Invitrogen #PA5-30055), 1:50/IP of anti-H3 antibody (Invitrogen, PA5-16183), 1:50/IP of anti-H4 antibody (Cell Signaling, #14149 S), 1 μg/IP of anti-H3K27me3

(Diagenode, #C15410195) and 1 μg/IP of anti-H3K36me3 (Diagenode, #C15410192) on a rotator overnight at 4 °C. Dependent on the added amount of antibody, the amount of Protein A magnetic beads (ThermoFisher Scientific #10001D) was adjusted (e.g., for 1–2 μg of antibody/IP = 15 μL of beads) and blocked overnight with 0.1% [w/v] bovine serum albumin in RIPA buffer. On the following day, beads were added to the IP samples for 2 h on a rotator at 4 °C to capture the antibody-bound fragments. The immunoprecipitated chromatin was subsequently washed twice with 150 μL each of ice-cold buffers RIPA-LS, RIPA-HS (10 mM Tris-HCl pH 8.0, 50 0 mM NaCl, 1 mM EDTA pH 8.0, 0.1% [w/v] SDS, 0.1% [v/v] sodium deoxycholate, 1% [v/v] Triton X-100), RIPA-LiCl (10 mM Tris-HCl pH 8.0, 250 mM LiCl, 1 mM EDTA pH 8.0, 0.5% [w/v] sodium deoxycholate, 0.5% [v/v] Nonidet P-40) and 10 mM Tris pH 8.0 containing protease inhibitors. Beads were washed once more with ice-cold 10 mM Tris pH 8.0, lacking inhibitors, and transferred into new tubes.

Beads were resuspended in 25 μL of the tagmentation reaction mix (Nextera DNA Sample Prep Kit, Illumina) containing 5 μL of 5× Tagmentation buffer, 1 μL of Tagment DNA enzyme, topped up with $H_2O$ to the final volume and incubated at 37 °C for 10 min in a thermocycler. Beads were mixed after 5 min by gentle pipetting. To inactivate the Tn5 enzyme, 150 μL of ice-cold RIPA-LS was added to the tagmentation reaction. Beads were washed twice with 150 μL of RIPA-LS and 1x Tris-EDTA and subjected to de-crosslinking by adding 100 μL ChIPmentation elution buffer (160 mM NaCl, 40 μg/mL RNase A (Sigma-Aldrich #R4642), 1× Tris-EDTA (Sigma #T9285) and incubating for 1 h at 37 °C followed by overnight shaking at 65 °C. The next day, 4 mM EDTA and 200 μg/mL Proteinase K (Roche, #03115828001) were added, and samples were incubated for another 2 h at 45 °C with 1000 rpm shaking. The supernatant was transferred into a new tube, and another 100 μL of ChIPmentation elution buffer was added for another hour at 45 °C with 1000 rpm shaking. DNA was isolated with MinElute PCR Purification Kit (Qiagen #28004) and eluted in 21 μL of $H_2O$.

DNA for the final library was prepared with 25 μL NEBNext Ultra II Q5 Master Mix, 3.75 μL IDT custom primer i5_n_x (10 μM); 3.75 μl IDT custom primer i7_n_x (10 μM) (see Supplementary Data 3); 3.75 μL $H_2O$ and 13.75 μL ChIPmentation DNA. The Cq value obtained from the library quantification, rounded up to the nearest integer plus one additional cycle, was used to amplify the rest of the ChIPmentation DNA. Library qualities were verified by High Sensitivity DNA Analysis on the Bioanalyzer 2100 (Agilent) before performing sequencing on NextSeq 500 (paired-end 35 bp reads) at the Core Unit Systemmedizin, Würzburg, Germany (samples without PAA) or DNBSEQ-G400 2x100bp in BGI, Hong Kong, China (samples with PAA). All samples were sequenced at equimolar ratios.

### Statistics and reproducibility
All experiments included at least two independent biological replicates.

### Read alignment
Quality control on sequencing reads was performed using fastQC[63]. Sequencing reads for ATAC-seq, RNA-seq, 4sU-seq, and ChIPmentation was mapped against (i) the human genome (GRCh37/hg19), (ii) human rRNA sequences and (iii) the *HSV-1* genome (*HSV-1* strain 17, GenBank accession code: JN555585, only for *HSV-1* infection data) using ContextMap v2.7.9[64] (using BWA as short read aligner[65] and allowing a maximum indel size of 3 and at most 5 mismatches). For the two repeat regions in the *HSV-1* genome, only one copy each was retained, excluding nucleotides 1–9213 and 145,590–152,222.

### Quality control and peak calling
Statistics on the numbers of mapped reads and reads mapped to human and *HSV-1* genomes were determined with samtools[66].

Promoter/transcript body (PT) scores were determined with ATACseqQC[67]. For peak calling in ATAC-seq and ChIPmentation data, BAM files with mapped reads were converted to BED format using BEDTools[68], and peaks were determined from these BED files using F-Seq with default parameters[69]. The fraction of reads in peaks (FRiP) was calculated with featureCounts[70] using identified peaks as an annotation. Annotation of peaks relative to genes was performed using ChIPseeker[71]. For ATAC-seq data, peaks identified to be in dOCRs were also additionally assigned to the downstream category.

### Analysis of open chromatin regions
dOCR length for a gene was calculated from OCRs (= peaks in ATAC-seq data) as previously described[5]. In brief, dOCRs were assigned to each gene in the following way. First, all OCRs overlapping with the 10 kb downstream of a gene were assigned to this gene. Second, OCRs starting at most 5 kb downstream of the so far most downstream OCR of a gene were also assigned to this gene. This was performed iteratively until no more OCRs could be assigned. Here, individual OCRs could be assigned to multiple genes. dOCR length of a gene was then calculated as the total genomic length downstream of this gene covered by OCRs assigned to the gene. Similarly, OCR length in gene bodies was calculated as the total genomic length of the gene bodies covered by OCRs.

### Quantification of downstream transcriptional activity and read-through
The number of read fragments per gene or in downstream regions was determined from the mapped RNA-seq or 4sU-seq reads in a strand-specific manner using featureCounts[70] and gene annotations from Ensembl (version 87 for GRCh37). For genes, all read pairs (= fragments) overlapping exonic regions on the corresponding strand by ≥25 bp were counted for the corresponding gene. For downstream regions, all fragments overlapping the 5 kb downstream of the gene 3'end were counted. Gene expression and downstream transcriptional activity were quantified in terms of fragments per kilobase of exons per million mapped reads (FPKM) and averaged between replicates. Only reads mapped to the human genome were counted for the total number of mapped reads for FPKM calculation. The percentage of read-through was calculated as previously described[5]. In brief: First, the percentage of transcription downstream of a gene was calculated separately for each replicate as the percentage of downstream transcription = 100 × (FPKM in 5 kb downstream of gene)/(gene FPKM). The percentage of downstream transcription was averaged between replicates, and the percentage of read-through was calculated as the percentage of downstream transcription in infected cells −the percentage of downstream transcription in uninfected or untreated cells. Negative values were set to 0.

### Metagene analyses
Metagene analyses were performed as previously described[72] using the software developed for this previous publication. For each gene, the regions −3 kb to +3 kb of the TSS were divided into 250 bp bins, the regions −5 kb to +100 kb of the TTS into 500 bp bins, and the remainder of the gene body (+3 kb of TSS to −5 kb of TTS) into 100 bins of variable length in order to compare genes with different lengths. For each bin, the average coverage per genome position was calculated and normalized to a total sum of 1. Metagene curves for each replicate were created by averaging results for corresponding bins across all genes considered, and metagene plots show the average metagene curves across replicates. To determine the statistical significance of differences between average metagene curves for two conditions, paired Wilcoxon signed rank tests were performed for each bin, comparing normalized coverage values for each gene for this bin between the two conditions. *P*-values were adjusted for multiple testing with the Bonferroni method across all bins within

each subfigure and are color-coded in the bottom track of subfigures: red = adj. $P$-value $\leq 10^{-5}$; orange = adj. $P$-value $\leq 10^{-3}$; yellow: adj. $P$-value $\leq 0.05$.

### Genome-wide differential analysis on histones and histone modifications

Differential analysis was performed with edgeR[73] on read counts determined with featureCounts for promoter regions (TSS ± 1.5 kb), gene bodies (TSS + 1.5 kb to TTS), and downstream regions (TTS to TTS + 25 kb) for all protein-coding and lincRNA genes in the Ensembl gene annotation. $P$-values were adjusted for multiple testing using the method by Benjamini and Hochberg. Distributions of log2 fold-changes for genes in individual clusters and specific types of genomic regions (promoter, gene bodies, or downstream regions) were then compared either against all other protein-coding and lincRNA genes or against all other genes included in the dOCR analysis using two-sided Wilcoxon rank sum tests. $P$-values for Wilcoxon tests were adjusted for multiple testing with the Bonferroni method separately for each histone or histone modification mark and infection.

### Data plotting and statistical analysis

All figures apart from Supplementary Fig. 8a–d and 12a–d were created in R[74] using the Bioconductor package Gviz[75] and the R packages gplots (for heatmaps) and ggplots (for all other figures). All statistical analyses were performed in R using the wilcox.test, fisher.test, and lm functions. Hierarchical clustering analysis was performed using the heatmap.2 function according to Euclidean distances and Ward's clustering criterion. Boxplots were created with default parameters, and linear regression analysis in scatter plots was performed using the lm function.

### Reporting summary

Further information on research design is available in the Nature Portfolio Reporting Summary linked to this article.

## Data availability

All sequencing data have been deposited at Gene Expression Omnibus (GEO) under accession code GSE185241 and GSE185239. Source data are provided in this paper. Genome sequences are available from UCSC for GRCh37/hg19 and GenBank for *HSV-1* strain 17 (accession JN555585), Ensembl annotations are available at https://www.ensembl.org/index.html. Source data are provided in this paper.

## Code availability

Workflows for read alignment and calculating dOCR lengths, downstream FPKM, and downstream transcriptional activity for the workflow management software Watchdog[76,77] are available at the Watchdog workflow repository (https://github.com/watchdog-wms/watchdog-wms-workflows, workflows: RNA_DifferentialGeneExpression, dOCRCalculation, Readthrough_Calculation). Watchdog, including installation instructions, is available at https://github.com/klugem/watchdog. Watchdog modules used in the workflows are available in the Watchdog module repository (https://github.com/watchdog-wms/watchdog-wms-modules). R scripts for creating figures are available at https://doi.org/10.5281/zenodo.7853167.

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

## Acknowledgements

This work was supported by the European Research Council (ERC-2016-CoG 721016—HERPES and ERC-2021-CoG 101041177—DecipherHSV to L.D.), the Deutsche Forschungsgemeinschaft (DO1275/6-1 to L.D., FR2938/9-1 to C.C.F. and FR2938/11-1 in the framework of the Research Unit FOR5200 DEEP-DV (443644894) to C.C.F.). A.W.W. was the recipient of a generous grant from the Alexander von Humboldt Foundation and the German Federal Foreign Office.

## Author contributions

L.Dj., T.He., C.C.F., and L.D. designed the experiments. L.Dj., T.He., A.M., K.W., A.G., E.W., and A.W.W. performed the experiments, and L.Dj., K.R., T.Ha., T.He., C.C.F., and L.D. analyzed the data. K.R., E.W., M.K., C.S.J., F.E., and C.C.F. performed computational analyses. L.Dj., T.He., A.W.W., C.C.F., and L.D. drafted the article. All authors contributed to revising the article draft.

## Funding

## Competing interests

The authors declare no competing interests.
