## [Peer Review File · Nature Communications]

The HSV-1 ICP22 protein selectively impairs histone repositioning upon Pol II transcription downstream of genesREVIEWER COMMENTS

Reviewer #1 (Remarks to the Author):

This manuscript shows that the HSV-1 protein ICP22 promotes the formation of open regions of chromatin that occur downstream of genes in response to HSV-1 infection (dOCRs). This adds new insight into the myriad of ways HSV-1 induces changes on the host genome during infection. The work is timely and moves the field forward, following prior publications from this group that investigated other aspects of disrupted termination of transcription during infection or cell stress.

Although this manuscript shows compelling links between virus-induced dOCRs, ICP22, and chromatin remodeling, the authors' conclusions need additional support. The following issues need to be addressed, some of which require new analyses or experiments. Consequently, as it stands now, the model presented is an over-interpretation of the data shown.

1. The authors used RNA-seq reads in the region 5kb downstream of genes as a metric to evaluate transcription across dOCRs that extend tens of thousands of kilobases downstream (scatter plots in Figures 2 and 3; Sup Fig 2 and 3). RNA-seq reads across the entire length of the dOCR need to be evaluated if the authors are to conclude that dOCR formation indeed depends on transcriptional activity through that region. Given their model that dOCRs are caused by impaired histone re-positioning in the wake of Pol II, showing that RNA is produced across these regions (or even better, Pol II ChIP-seq) is important.
2. There are often genes located within the regions the authors define as dOCRs (for example in Figure 3b, there are 2 other genes in the region defined as the dOCR for HNRNPA2B1). Therefore, it is possible the open chromatin arises from transcription of the genes embedded within the dOCR, rather than from DoTT as the authors propose. This needs to be addressed with additional analyses. For example, analyses with the dOCRs for genes that are "isolated" in the genome and do not contain other genes within the downstream region encompassed by a dOCR. Alternatively, the continuity of RNA-seq reads across the dOCR could be evaluated - do the reads show increases and decreases at the beginnings/ends of the embedded genes, which would indicate transcription of the embedded genes is also happening in conjunction with DoTT?
3. The 4sU-seq data from a prior publication (Figures 1c and 1d) are not relevant to most of the infection conditions used in panels 1a and 1b, which include strain F, longer time points of infection, and PAA. The prior 4sU-seq data should be removed and replaced with data collected under the experimental conditions for this manuscript. Ideally this would be new 4sU-seq data, however, the total RNA-seq data that the authors already have could be used. That said, why was total RNA-seq performed in this manuscript instead of 4sU-seq? RNA degradation/stability is impacted by HSV-1 infection, thus changes in total RNA-seq are not necessarily a proxy for changes in transcription. Can the authors comment on the level of ongoing cellular transcription under the infection conditions used here?

4. Figure 3 lacks important controls. For the ICP22 cells, please show +Dox, -salt data to support the conclusion that Dox-induced ICP22 expression does not induce dOCRs in the absence of read-through transcription. In addition, -Dox, -salt data need to be shown to support the conclusion that salt itself does not induce dOCRs. Then in the ICP22+ICP27 cells, we cannot know that ICP22 is causing the induction of dOCRs as opposed to ICP27; the impact of ICP27 alone on dOCRs needs to be shown as a control.

5. More evidence is needed to support the conclusion that the increase in dOCRs due to knockdown of FACT is specific for DoTT. The data clearly show that knockdown of FACT increases chromatin accessibility across gene bodies, and there are genes embedded within the regions defined as dOCRs. Thus, it is possible that the increase in dOCRs in response to FACT knockdown can be attributed to increased chromatin accessibility within gene bodies as opposed to DoTT. Obtaining RNA-seq data in the presence of FACT knockdown is needed, followed by analyses to distinguish DoTT from transcription of genes embedded within dOCRs.

Minor revisions

1. From Figure 1b the authors conclude that 6 clusters do not show induction of dOCRs upon infection, which represents the majority of the genes. The authors need to discuss how this mixed response/selectivity might arise in light of their model for dOCR formation.

2. For the sequencing data collected before and after infection, please add a table in the supplement showing the number of sequencing reads that map to the host genome and the viral genome. This will allow the reader to evaluate the extent to which mutant viruses and PAA impact viral replication, and whether the FPKM values change as expected during infection.

3. Please include in the methods a description of how the similarity between experimental replicates was evaluated or statistically considered. Looking at Figure 1b, many of the replicates do not look very similar.

4. A paper was recently published showing that ICP22 associates with FACT and represses Pol II elongation. In the Discussion, please reconcile their findings with your model. (Isa, N.F. et al, Vaccines (2021) doi.org/10.3390/vaccines9101054)

5. The RNA-seq data needs to be displayed along with the ATAC-seq data for the representative regions of the genome shown in Figs 3b, 5d, and Sup Fig 1a.

6. Please describe the difference between strain 17 and strain F. Could the differences explain why their dOCR responses are different?

7. Please make sure all figure legends clearly state which set of genes are being plotted in which panels and the statistical test used.

8. For Fig 3c, the -Dox data need to also be shown in the supplement, much like the data in panel 3d.

Reviewer #2 (Remarks to the Author):

The submitted manuscript describes a series of transcriptomics and epigenomics experiments evaluating the roles of HSV-1 proteins in inducing downstream open chromatin regions (dOCR). In brief, the generation of dOCRs does not require high levels of late proteins, it is enhanced in the presence of PAA, and it is inhibited by deletions of ICP22 and to lesser extent of IC27. Expression of ICP22 does not appear sufficient on its own to reproduce the effect, but it does in addition to a secondary stress. The authors conclude that dOCR induction is a consequence of deficient histone deposition in the wake of RNA Pol II transcription through a depletion of FACT by sequestration into replication compartments. However, FACT deletion did not restore the phenotype during infections with ICP22 null virus, and although it enhanced it during infections with wild-type virus, it also resulted in apparent histone depletion in the bodies of some subsets of genes. Another limitation to the model proposed is that histone occupancy in the relevant areas is not evaluated under conditions in which RNA Pol III transcription is inhibited. The proposed model thus remains speculative with limited experimental support at this time.

Specific issues.

The major limitation lies in the model proposed. The model proposes that ICP22 mediates dOCR by its recruitment of the FACT subunits into the replication compartments (worded as "functional inhibition", and fully described in page 11, last paragraph), away from the cellular chromatin preventing the proper histone deposition in the wake of RNA Pol II. However, several pieces of evidence are difficult to reconcile with this model. Firstly, ectopic expression of ICP22 in the absence of replication compartments still can induce dOCR under conditions of transcriptional stress. Perhaps the model proposes that the requirement for a transcriptional stress to phenocopy the dOCR phenotype is a consequence of the impossibility of recruitment of FACT into the replication compartments, but this possibility is not actually discussed; neither is it discussed how ICP22 would functionally inhibit FACT in the absence of replication compartments. ICP22 is also speculated to induce the increase in dOCR by inducing modifications of the RNA Pol III phosphorylation status, but this hypothesis is not tested experimentally either. In brief, the proposed mechanism is not obviously consistent with the presented results.

The criteria used for the cut offs to segregate clusters 1-9 by dOCR length are not discussed, which hampers the readers' ability to evaluate these analyses. Furthermore, this clustering analysis shows clusters in which the dOCR are present in uninfected and increase in infected cells, and clusters in which they increase in PAA-treated infections, as discussed, but also clusters in which the dOCR are present in uninfected cells, decrease in infected cells, and are

enriched again in infected cells treated with PAA. Some of these clusters (for example a subset in cluster 3) appear to include several genes. It is unclear how does the transcription and dOCR patterns of the genes in these clusters fit into the proposed model.

The box plots presented in figures 1 c-d, and supplementary figure S1c are not described and the y-axes in some instances are truncated at too low of a value to observe the actual dispersion of all datasets. The description of the box plots is somewhat unclear. For example, when discussing the % read-through presented in figure 1c, it is mentioned that Cluster 2 had the highest percentage. However, the box plots overlap to great extent and no statistical analyses are provided to support this statement. Likewise, although the difference between cluster 5 and all other clusters in figure 1d is visible, no statistical evaluation is provided to support this conclusion. The same criticism applies to Supp Fig 1c, with regards to the conclusions about transcription levels in clusters 2, 5, and 6.

The biological meaning of the correlation between the frequency of read through transcription (downstream FPKM) and dOCR length is not clear. According to the statistics provided, these two parameters appear to be highly correlated in wild type infections and some other circumstances. However, the actual plot dispersion and the slopes show limited obvious correlation between the two parameters in many instances, with the residuals appearing to be larger than the regression. This reviewer could not find the statistical tests used to calculate the p values, or whether the statistical analyses performed evaluated whether the slope was different from 0. The statistical analyses must be fully described, and the authors may want to expand on their analyses of these correlations to better convey their potential biological meaning, in particular when they have such a large dispersion. In these plots, there is also a striking lack of downstream FPKM in between certain low values, which may well be artifactual. The authors should discuss the reasons for this abnormal distribution, or if it were truly artifactual, this artifact should somewhat be accounted for in the analyses.

The meta genome analyses of the levels of histone deposition presented in figure 4 show the discussed relative depletion past the transcription termination sites, but also present some apparent enrichments in other regions. It is unclear how these differences fit into the general model proposed. These meta gene analyses show only genes in cluster 5, which as discussed are not representative of all genes. It is also unclear whether the analyses of the modified histones are corrected by histone occupancy.

The analysis of the correlation between dORC with and without induction in figure 5 only discusses the genes that have longer dORC upon induction, omitting the genes with decreased dOCR upon induction. Moreover, no statistical analyses are performed to evaluate whether the discussed differences between genes in cluster 5 and other clusters are significant, or whether any cluster departs significantly from the average.

The authors state that "that the viral genome remains largely unchromatinized during productive infection", citing two papers presenting evidence that is consistent with this

interpretation (and also with others) without discussing the many other publications using ChIP and other approaches have consistently presented evidence that it is not consistent with this conclusion. A more balanced discussion would be highly desirable.

The methods should be expanded to fully describe the analyses, statistics and data plotting in more detail.

Reviewer #3 (Remarks to the Author):

HSV-1 disrupts transcription termination causing downstream transcription and chromatin changes to a subset of cellular genes. The viral factors and their interactions with host transcriptional mechanisms remains unresolved. This study utilizes ATAC-seq and RNA-seq/4sU-seq analysis to examine the chromatin accessibility associated with downstream transcription in primary human fibroblasts following acute infection with different HSV-1 strains and treatment with viral replication inhibitor. The results classify downstream open chromatin regions (dOCRs) that exhibit read-through transcription, and identify ICP22 and FACT as required for dOCR production. Authors conclude that ICP22 protein regulates FACT histone chaperone to alter Pol II composition and histone nucleosome repositioning.

This study provides significant insight into the chromatin and transcriptional mechanisms associated with HSV-1 infection. Overall, the technical detail is satisfactorily described and appropriately presented, and the conclusions are supported by the results. However, there are few major critiques such as a request to provide additional data analysis for quality control as well as an analysis of the global HSV-1-induced chromatin changes beyond what is presented. In addition, there are a few minor grammatical and scientific critiques noted that should be addressed.

Supplemental ATAC-seq, Omni-ATAC-seq and ChIPmentation information and quality control should be provided. i.e., table of that provides total number of raw reads, aligned reads (both human and HSV-1), number of called peaks (ATAC-seq, ChIPmentation) and other assay-specific QC measures; promoter scores, chrM alignment rates, FRiP, etc. This analysis will support several points raised in the manuscript. For example, in lines 132-134 authors discuss PAA treatment of infected cells causing a much higher percentage of cellular reads in the ATAC-seq data due to PAA. This is unclear, are the authors referring to reads that align unique to the human genome? Are there similar read depth in the libraries that assess dOCR length differences, and how does PAA impact chrM reads in ATAC-seq experiments? These QC results should support several statements including those from lines 134-135, 241-243, 282-283, etc.

Similarly, authors conclude that dOR represents majority of chromatin changes, however a statistical differential accessibility analysis was not performed, only dOCR length. Does the majority of differential accessibility with infection fall downstream of TTS regions (dOCRs)? Does virus infection significantly alter OCR outside of dOCR?

It is unclear how the 4,612 genes (line 125) that exhibit read-through transcription were identified. Were they determined by infection of HFF cells with one or both viral strains and with PAA treatment (i.e. all conditions)? Or were these defined by previous analysis and are representative of genes that are susceptible to this phenomenon? The description of host genes with dOCR should be more clearly described. Similarly, it is unclear from results text and figure legend that among the 4,612 dOCRs, only ~500 genes are shown that 'reach at least the length indicated on the x-axis in mock and WT HSV-1 infection'. However, the minimum length indicated on the x-axis is 0 and not all 4,612 genes are shown (presumably because ~4,100 dOCRs are < ~10kb?).

Line 125 discusses filtering genes for read-in transcription originating from read-through transcription of an upstream gene. However, the concept of read-in is not obvious and should be defined in the background to avoid confusion. Also, the log₁₀ dOCR length scale of the Figure 1B doesn't seem to resolve the lengths consistent in Figure 1A. Consider changing the colors of the scale and the Figure 1B color bar and legend should specify the units of length for clarity (bp or kb?).

Line 190 suggests only one viral gene is responsible for dOCR induction.

Line 212 needs to reference Fig. 2c

Line 223-225, authors state that ICP27 is responsible for the majority of HSV-1 DoTT, but from the results presented this is not clear. Please clearly reference this data or the appropriate citation.

Authors state changes in genome-wide occupancy of histone occupancy, but the metagene analysis referenced in figure 4 is not a genome-wide analysis. An analysis of differential enrichment should be performed and statistically significant changes should be mapped with respect to gene bodies or dOCRs to support conclusions.

Line 271 please clarify what is meant by partly statistically significant

Figure 2a legend should reference figure 1b and state the number of OCRs as cluster 5 alone is not clear. Also, it should be also be clarified in the figure legend that the difference between Fig 2a and 2b is strains 17 and F, respectively. Finally, the legend states 'Experiments for (b) were performed with and without PAA treatment', but it appears PAA is in both Fig2a and 2b.

The methods section called 'RNA-seq controls' discusses confirmation of read-through transcription under the same experimental samples for OCR analysis, yet previously published data from alternative experiments was used to generate the figures. Supportive data indicating that read-through transcription was induced in these conditions should be provided, as well as supplemental data that indicates similar to correlations to expression in

these experiments.

Lines 59-60 states open chromatin is only observed around gene promoters and gene bodies. This is not an entirely accurate statement as accessibility can also be used to map intergenic regions.

Lines 70-72 are incomplete sentences.

REVIEWER COMMENTS

Reviewer #1 (Remarks to the Author):

This manuscript shows that the HSV-1 protein ICP22 promotes the formation of open regions of chromatin that occur downstream of genes in response to HSV-1 infection (dOCRs). This adds new insight into the myriad of ways HSV-1 induces changes on the host genome during infection. The work is timely and moves the field forward, following prior publications from this group that investigated other aspects of disrupted termination of transcription during infection or cell stress.

We would like to thank the reviewer for the thorough and positive evaluation of our work.

Although this manuscript shows compelling links between virus-induced dOCRs, ICP22, and chromatin remodeling, the authors' conclusions need additional support. The following issues need to be addressed, some of which require new analyses or experiments. Consequently, as it stands now, the model presented is an over-interpretation of the data shown.

1. The authors used RNA-seq reads in the region 5kb downstream of genes as a metric to evaluate transcription across dOCRs that extend tens of thousands of kilobases downstream (scatter plots in Figures 2 and 3; Sup Fig 2 and 3). RNA-seq reads across the entire length of the dOCR need to be evaluated if the authors are to conclude that dOCR formation indeed depends on transcriptional activity through that region. Given their model that dOCRs are caused by impaired histone re-positioning in the wake of Pol II, showing that RNA is produced across these regions (or even better, Pol II ChIP-seq) is important.

This is an excellent suggestion to strengthen our findings. We have now visualized read coverage in our total RNA-seq and ATAC-seq data in 1kb windows across whole dOCR regions for genes with strong induction of dOCRs, i.e. genes in Cluster 5. We show the results of this analysis for (i) WT HSV-1 infection with and without PAA treatment and mock infection (new **Sup. Fig. 3**), (ii) ectopic co-expression of ICP22 and ICP27 (**Sup. Fig. 7g**) and (iii) SSRP1 knock-down (new **Sup. Fig. 13**). This confirmed extensive transcription throughout dOCRs in HSV-1 infection and cells co-expressing ICP22 and ICP27, but not in mock infection. Moreover, it further confirms that lower transcriptional activity in WT infection without PAA treatment explains lower levels of dOCRs compared to WT infection with PAA treatment (**Sup. Fig. 3**). In addition, we visualize RNA-seq read coverage in Δ ICP22 infection on dOCRs identified in WT strain F infection (**Sup. Fig. 5c,d**). Δ ICP22 infection showed only slightly reduced transcription downstream of genes compared to WT infection. Despite extensive transcription downstream of genes no dOCRs were induced. Thus, the failure of Δ ICP22 infection to induce dOCRs does not result from a lack of transcription downstream of genes.

2. There are often genes located within the regions the authors define as dOCRs (for example in Figure 3b, there are 2 other genes in the region defined as the dOCR for HNRNPA2B1). Therefore, it is possible the open chromatin arises from transcription of the genes embedded within the dOCR, rather than from DoTT as the authors propose. This needs to be addressed with additional analyses. For example, analyses with the dOCRs for genes that are "isolated" in the genome and do not contain other genes within the downstream region encompassed by a dOCR. Alternatively, the continuity of RNA-seq reads across the dOCR could be evaluated - do the reads show increases and decreases at the beginnings/ends of the embedded genes, which would indicate transcription of the embedded genes is also happening in conjunction with DoTT?

Continuity of RNA-seq reads throughout the whole dOCR regions is now shown in the analysis performed for comment #1. We furthermore performed this analysis – as suggested by the reviewer – selectively for Cluster 5 genes with no known protein-coding gene or lincRNA within the first 50kb downstream of their gene 3' end (new **Sup. Fig. 4**). This confirmed our observations made for all genes with strong dOCR induction (Cluster 5) and shows that dOCRs do not arise due to transcription of genes embedded in these regions.

We also performed the same analysis for the data from ectopic ICP22 and ICP27 co-expression (new **Sup. Fig. 7h**).

3. The 4sU-seq data from a prior publication (Figures 1c and 1d) are not relevant to most of the infection conditions used in panels 1a and 1b, which include strain F, longer time points of infection, and PAA. The prior 4sU-seq data should be removed and replaced with data collected under the experimental conditions for this manuscript. Ideally this would be new 4sU-seq data, however, the total RNA-seq data that the authors already have could be used.

We fully agree with the referee that the 4sU-seq data are not directly corresponding to several of the infection conditions included in the manuscript and highlighted by the reviewer, i.e., WT strain F with longer time points of infection and PAA treatment. However, the 4sU-seq time-course data provide a detailed picture of the kinetics of read-through transcription in HSV-1 infection. They revealed that absolute levels of downstream transcriptional activity (FPKM in the 5kb downstream of the gene 3' end), rather than relative read-through levels, represent the key factor that defines the extent of dOCR formation.

Infection with different virus strains or null mutants can impact dOCR induction in two different ways: It may either alter transcription levels downstream of the respective genes and thus indirectly affect dOCR induction, or it may directly abolish dOCR formation due to the lack of the responsible viral factor. We would like to present the data based on how we approached this question for the mutant viruses, namely, by first looking for dOCRs in regions which showed dOCR formation in WT infection, and then look at viral mutants that no longer induced dOCRs to check whether there is still transcription in the respective regions downstream of genes.

Accordingly, looking for dOCR formation in Δ ICP0, Δ ICP22, Δ ICP27 and Δ vhs infection excluded ICP0, ICP27 and vhs as viral factor(s) directly responsible for dOCR formation but indicated that ICP22 is required. To confirm this, we analyzed the total RNA samples from the relevant conditions, which were obtained for the same samples as used for ATAC-seq. This included WT strain F as the parental strain of the Δ ICP22 mutant in addition to this null mutant, longer time points of infection and +/-PAA treatment (see also response to the next comment).

We have now also included the analysis of read-through, downstream transcriptional activity and gene expression in these total RNA-seq data in the manuscript (new **Sup. Fig. 2**). This confirms the results on the 4sU-seq data. As the 4sU-seq data visualize the kinetics through-out the first 8 h of infection more nicely and are fully consistent with the new total RNA-seq data, we kept Figures 1c and 1d and included the analysis on the total RNA-seq data as new **Sup. Fig. 2**

That said, why was total RNA-seq performed in this manuscript instead of 4sU-seq? RNA degradation/stability is impacted by HSV-1 infection, thus changes in total RNA-seq are not necessarily a proxy for changes in transcription. Can the authors comment on the level of ongoing cellular transcription under the infection conditions used here?

While 4sU-seq data indeed provide a more direct measure of transcriptional activity on genes during infection, generally no or very little transcriptional activity is present downstream of genes in uninfected cells. Therefore, virtually all total RNA-seq reads mapping to genomic regions downstream of genes arise from transcription that occurred after infection. Thus, total RNA-seq is fully suitable to both confirm and quantitatively assess transcriptional activity downstream of genes during infection. Analysis of total RNA instead of 4sU-RNA only requires a greater sequencing depth to obtain the same read coverage downstream of genes. Results from 4sU-seq and total RNA-seq data are thus highly consistent for cellular transcription downstream of genes and showed that absolute levels of downstream transcriptional activity are the best predictor for the induction of dOCRs both for different strains and time-points of WT infection and at individual gene level. Therefore, total RNA-seq is suitable (and much less work-intensive to obtain) for identifying conditions/null mutants in which dOCRs are not induced despite the presence of downstream transcriptional activity. In contrast to 4sU-seq analysis, which requires >30 μ g of RNA as starting material, total RNA-seq only requires <1 μ g of input

sample and has a reduced risk for drop-outs. We thus always freeze samples for total RNA-seq in ATAC-seq even if we do not directly perform RNA-seq. The new RNA-seq samples for FACT knock-down (see response to comment #5) were indeed taken from frozen RNA obtained together with the corresponding ATAC-seq experiments. They thus directly match the respective ATAC-seq data in the paper.

4. Figure 3 lacks important controls. For the ICP22 cells, please show +Dox, -salt data to support the conclusion that Dox-induced ICP22 expression does not induce dOCRs in the absence of read-through transcription. In addition, -Dox, -salt data need to be shown to support the conclusion that salt itself does not induce dOCRs. Then in the ICP22+ICP27 cells, we cannot know that ICP22 is causing the induction of dOCRs as opposed to ICP27; the impact of ICP27 alone on dOCRs needs to be shown as a control.

We agree with the reviewer that these controls were missing. This was due to dropouts to some of the respective samples in the sequencing run due to problems in the library preparation which were only picked up after sequencing. We now repeated the experiment and performed parallel Omni-ATAC- and RNA-seq for T-HFs-ICP22 cells with and without Dox exposure as well as cells that express ICP27 upon Dox exposure (T-HFs-ICP27 cells) and T-HFs-ICP22 cells with and without salt stress. Results are shown in new **Sup. Fig. 7e,f**. This confirmed that Dox-induced expression of either ICP22 or ICP27 alone is not sufficient for dOCR induction. While salt stress without Dox exposure did not induce dOCRs in T-HFs-ICP22 cells in our first experiment, we did see some dOCR induction in the second control experiment for salt stress even without Dox exposure. This was explained by a greater extent of leaky ICP22 expression in absence of Dox. Since we previously observed no dOCR induction in primary HFFF cells upon salt and heat stress, leaky ICP22 expression in absence of Dox, which was confirmed in the RNA-seq data, explains low-level dOCR formation in T-HFs-ICP22 cells exposed to salt stress. Nevertheless, in both experiments, Dox-induced ICP22 expression led to increased dOCR induction.

5. More evidence is needed to support the conclusion that the increase in dOCRs due to knockdown of FACT is specific for DoTT. The data clearly show that knockdown of FACT increases chromatin accessibility across gene bodies, and there are genes embedded within the regions defined as dOCRs. Thus, it is possible that the increase in dOCRs in response to FACT knockdown can be attributed to increased chromatin accessibility within gene bodies as opposed to DoTT. Obtaining RNA-seq data in the presence of FACT knockdown is needed, followed by analyses to distinguish DoTT from transcription of genes embedded within dOCRs.

The reviewer raises an important point here. We thus repeated the analysis on dOCR length shown in **Fig. 5a,b** selectively for Cluster 5 genes with no known protein-coding or lincRNA genes within 50kb downstream of the gene 3'end. This also showed dOCR induction in HSV-1 infection upon FACT depletion (new **Sup. Fig. 12f**). As the reviewer suggested, we furthermore performed RNA-seq analysis of total RNA samples from the respective experiments (same samples), which we had stored at -80°C for future analysis (see response to comment #3). New **Sup. Fig. 13** now shows RNA-seq and ATAC-seq read coverage on dOCR regions after FACT knock-down in HSV-1 infection both for all Cluster 5 genes (**Sup. Fig. 13a-d**) and selectively for genes with no known protein-coding or lincRNA genes within 50kb downstream of the gene 3'end (**Sup. Fig. 13e-h**). This demonstrates both the presence of transcriptional activity throughout dOCRs as well as the presence of ATAC-seq reads throughout dOCRs. This excludes that increases in dOCR induction in HSV-1 infection upon FACT knock-down are due to increased chromatin accessibility within gene bodies.

Minor revisions

1. From Figure 1b the authors conclude that 6 clusters do not show induction of dOCRs upon infection, which represents the majority of the genes. The authors need to discuss how this mixed response/selectivity might arise in light of their model for dOCR formation.

Our data shows that strong transcriptional activity downstream of genes upon Pol II read-through is required for dOCR induction, likely due to impaired histone repositioning. Genes that exhibit dOCR induction are either highly expressed (Cluster 6) or show strong read-through (relative to their gene expression) (Cluster 2) or both (Cluster 5). For the genes in the remaining six clusters, which are neither particularly highly expressed nor show particular high read-through, Pol II transcription downstream of genes is thus not sufficiently high to induce dOCRs. In light of the HSV-1-induced transcriptional host shut-down, it may well be that the majority of host genes are no longer transcribed at all or at very low levels in the vast majority of cells beyond 4 h p.i. (see also PRO-seq studies from the Baines lab). Considering how efficiently HSV-1 shuts down host transcription throughout infection, it is thus not surprising that many genes do not show sufficient transcriptional activity downstream of genes at late times of infection for dOCR formation. Accordingly, salvage of cellular Pol II transcription by PAA treatment drastically increases dOCR formation. We now clarified this and included a more detailed discussion on this into the manuscript.

2. For the sequencing data collected before and after infection, please add a table in the supplement showing the number of sequencing reads that map to the host genome and the viral genome. This will allow the reader to evaluate the extent to which mutant viruses and PAA impact viral replication, and whether the FPKM values change as expected during infection.

We included the respective information in new Sup. Tab. 1 for the ATAC-seq data and Sup. Tab. 2 for ChIPmentation data.

3. Please include in the methods a description of how the similarity between experimental replicates was evaluated or statistically considered. Looking at Figure 1b, many of the replicates do not look very similar.

We now included a figure showing Spearman correlations (ρ) in dOCRs length between all replicates for mock and WT infection (new **Sup. Fig. 1b**). dOCR lengths were highly correlated between replicates and between similar conditions, in particular for conditions with strong induction of dOCRs, i.e., WT infection with PAA treatment (both strains) and WT strain 17 infection ($\rho > 0.8$). As expected, the correlation between biological replicates was weaker ($\rho > 0.64$) for conditions without dOCR induction (mock and mock + PAA) or less pronounced dOCR induction (WT strain F infection without PAA) due to the “noise” provided by chromatin accessibility arising from other sources.

4. A paper was recently published showing that ICP22 associates with FACT and represses Pol II elongation. In the Discussion, please reconcile their findings with your model. (Isa, N.F. et al, Vaccines (2021) doi.org/10.3390/vaccines9101054)

We would like to thank the reviewer for pointing this out. We now mention the respective paper in the introduction, results section and discussion.

5. The RNA-seq data needs to be displayed along with the ATAC-seq data for the representative regions of the genome shown in Figs 3b, 5d, and Sup Fig 1a.

We included the respective data in **Figs. 3b, 5d, Sup. Fig. 1a** and **Sup. Fig. 14**.

6. Please describe the difference between strain 17 and strain F. Could the differences explain why their dOCR responses are different?

While we observed a stronger induction of dOCRs upon infection with strain 17 than with strain F, we would like to stress that this is only a minor quantitative difference. As can be seen for the SRSF3 gene in **Sup. Fig. 1a**, both strains induce quite similar levels of dOCRs. dOCR formation in HSV-1 infection

depends on (i) ICP27-induced DoTT, (ii) ICP22-mediated effects presumably on FACT, and (iii) strain-specific differences in the kinetics of productive infection and thus transcriptional host shut-off for the cell type under study. ICP22 is identical between WT strains 17 and F (as now noted in the discussion), while ICP27 shows only two amino acid changes between WT strains 17 and F. We thus hypothesize that general properties of these two viruses that impact infection kinetics explain the observed differences. Since WT strain F also exhibits reduced read-through transcription consistent with attenuated infection compared to WT strain 17, the likely explanation for reduced dOCR induction in WT strain F infection is reduced transcriptional activity downstream of genes. As we feel that strain-specific differences are minor and not important for the main findings of the manuscript, we would like to avoid a more detailed discussion of the observed strain-specific differences. However, we do note reduced read-through transcription of WT strain F in the first results section as a likely explanation of reduced dOCR induction and now additionally highlight the connection between transcriptional activity downstream of genes and dOCR induction in the discussion.

7. Please make sure all figure legends clearly state which set of genes are being plotted in which panels and the statistical test used.

We adjusted the figure legends accordingly.

8. For Fig 3c, the -Dox data need to also be shown in the supplement, much like the data in panel 3d.

We now included the respective data into **Sup. Fig. 7j**.

Reviewer #2 (Remarks to the Author):

The submitted manuscript describes a series of transcriptomics and epigenomics experiments evaluating the roles of HSV-1 proteins in inducing downstream open chromatin regions (dOCR). In brief, the generation of dOCRs does not require high levels of late proteins, it is enhanced in the presence of PAA, and it is inhibited by deletions of ICP22 and to lesser extent of ICP27. Expression of ICP22 does not appear sufficient on its own to reproduce the effect, but it does in addition to a secondary stress. The authors conclude that dOCR induction is a consequence of deficient histone deposition in the wake of RNA Pol II transcription through a depletion of FACT by sequestration into replication compartments. However, FACT deletion did not restore the phenotype during infections with ICP22 null virus, and although it enhanced it during infections with wild-type virus, it also resulted in apparent histone depletion in the bodies of some subsets of genes. Another limitation to the model proposed is that histone occupancy in the relevant areas is not evaluate under conditions in which RNA PolIII transcription is inhibited. The proposed model thus remains speculative with limited experimental support at this time.

We would like to thank the reviewer for the detailed evaluation of our work highlighting important issues that we have now all addressed.

Specific issues.

The major limitation lies in the model proposed. The model proposes that ICP22 mediates dOCR by its recruitment of the FACT subunits into the replication compartments (worded as "functional inhibition", and fully described in page 11, last paragraph), away from the cellular chromatin preventing the proper histone deposition in the wake of RNA Pol II. However, several pieces of evidence are difficult to reconcile with this model. Firstly, ectopic expression of ICP22 in the absence of replication compartments still can induce dORC under conditions of transcriptional stress. Perhaps the model proposes that the requirement for a transcriptional stress to phenocopy the dORC phenotype is a consequence of the impossibility of recruitment of FACT into the replication compartments, but this possibility is not actually discussed; neither is it discussed how ICP22 would functionally inhibit FACT in the absence of replication compartments. ICP22 is also speculated to induce the increase in dOCR by inducing modifications of the RNA PolIII phosphorylation status, but this hypothesis is not tested experimentally either. In brief, the proposed mechanism is not obviously consistent with the presented results.

There seems to have been a misunderstanding here about the model we propose. As stated at the end of the introduction, "We propose a model in which functional inhibition of FACT by the viral ICP22 protein and allosteric changes in Pol II composition downstream of genes result in selectively impaired histone repositioning in the wake of Pol II." Apparently, the term "functional inhibition" may have been misleading. We revised the discussion on this to describe the proposed model more clearly.

Importantly, "strong dOCR induction in absence of viral DNA replication implies that deprivation of cellular factors by their recruitment to viral replication compartments **is not** responsible for induction of dOCRs" (see first paragraph of discussion). We hypothesize that ICP22 impairs recruitment of FACT to Pol II resulting in a "functional impairment" of FACT activity for actively transcribing Pol II (see 4th paragraph of discussion). FACT interacts with multiple Pol II-associated factors and is likely to be recruited to Pol II by more than one mechanism. We hypothesize that (*i*) ICP22 only interferes with one mode of FACT recruitment and that (*ii*) the allosteric changes in Pol II upon (partial) recognition of the poly(A) site abrogate other modes of FACT recruitment (i.e., possibly in the frame of a general loss of transcription elongation factors). This would explain why dOCRs in HSV-1 infection (and upon ectopic ICP22 expression) predominantly arise downstream of genes whereas knockdown of FACT increases chromatin accessibility both up- and downstream of poly(A) sites.

Besides interacting with FACT, one of the main effects of ICP22 is a global loss of serine 2 phosphorylation (Ser-2P) of the C-terminal domain of Pol II. The latter functions as a docking platform for cellular factors involved in transcription. We thus cannot exclude that ICP22 may interfere with

FACT activity in active Pol II transcription by its effects on Ser-2P. We have now sharpened the discussion on this.

Importantly, many genes show a concordant increase in chromatin accessibility both up- and downstream of their poly(A) site upon HSV-1 infection of cells depleted for FACT. It appears highly unlikely that two completely unrelated molecular mechanisms would result in such concordant alterations.

The criteria used for the cut offs to segregate clusters 1-9 by dOCR length are not discussed, which hampers the readers' ability to evaluate these analyses.

We now describe in the manuscript how the nine clusters were obtained from the hierarchical clustering. Specifically, “inspection of the clustering dendrogram and heatmap indicated at least three clusters with increasing dOCR lengths in infection (marked as Clusters 2, 5 and 6 in **Fig. 1b**). The cutoff on the dendrogram was thus chosen such that these clusters were obtained as separate clusters, resulting in a total of nine different gene clusters.”

It is important to note that these nine clusters are not set in stone but rather used only as a tool to identify the viral genes that are or are not involved in dOCR formation. While the susceptibility of a gene's poly(A) site to DoTT by HSV-1 is likely an inherent feature, dOCR induction also depends on the expression level of the respective gene in the cells under study. This will vary between different cell types and conditions. Validation of viral candidate genes (ICP22 and ICP27) responsible for dOCR formation was then performed based on genes with strong transcription downstream of genes in the respective experimental conditions. We now included a statement on this into the discussion.

Furthermore, this clustering analysis shows clusters in which the dOCR are present in uninfected and increase in infected cells, and clusters in which they increase in PAA- treated infections, as discussed, but also clusters in which the dOCR are present in uninfected cells, decrease in infected cells, and are enriched again in infected cells treated with PAA. Some of these clusters (for example a sub-set in cluster 3) appear to include several genes. It is unclear how does the transcription and dOCR patterns of the genes in these clusters fit into the proposed model.

It is important to note that ATAC-seq measures chromatin accessibility which may arise by multiple different mechanisms, only one of which is impaired histone repositioning in dOCR regions. Some level of biological “noise” is thus to be expected. Cluster 3 genes tend to be relatively highly expressed (comparably to Clusters 2 and 6). These genes tend to have some open chromatin downstream of genes already in mock infection, likely due to Pol II transcribing beyond poly(A) sites before transcription termination. Since transcriptional activity on host genes is reduced in WT infection (most strongly in 12 h p.i. WT infection) but again increased with PAA treatment, this suggests that loss of host transcriptional activity in WT infection without PAA treatment and reduced loss of transcriptional activity in WT + PAA treatment is responsible for this effect. This is also consistent with our observation that significant transcriptional activity downstream of gene is required to induce dOCRs. It should be noted that there may also be some increase in dOCR length in WT + PAA treatment for some genes in other clusters than Clusters 2, 5 and 6. However, since this was quite weak, we focused only on genes with clear evidence of dOCR induction, in particular Cluster 5. We expanded the discussion on the proposed model and how transcription and dOCR patterns of the genes in these clusters fit into the proposed model.

The box plots presented in figures 1 c-d, and supplementary figure S1c are not described and the y-axes in some instances are truncated at too low of a value to observe the actual dispersion of all datasets. The description of the box plots is somewhat unclear. For example, when discussing the % read-through presented in figure 1c, it is mentioned that Cluster 2 had the highest percentage. However, the box plots overlap to great extent and no statistical analyses are provided to support this statement. Likewise, although the difference between cluster 5 and all other clusters in figure 1d is visible, no statistical evaluation is provided to support this conclusion. The same criticism applies to Supp Fig 1c, with regards to the conclusions about transcription levels in clusters 2, 5, and 6.

Boxplots are now described in Figure legends of **Fig. 1c, d, Sup. Fig. 1c** and new **Sup. Figs. 2, 10** and **11**, i.e., boxes represent the range between the first and third quartiles for each condition. Black horizontal lines in boxes show the median. The ends of the whiskers (vertical lines) extend the box by 1.5 times the inter-quartile range. Data points outside this range (outliers) are shown as small circles.

Significance of differences between clusters were assessed with one-sided Wilcoxon rank sum tests. The p-values for these tests and a description of the statistical evaluation are now provided in the revised manuscript. Furthermore, **Fig. 1c, d** and **Sup. Fig. 1c** and new **Sup. Fig. 2** are now included with less truncation to allow assessing dispersion of datasets. We did nevertheless perform some truncation as otherwise presence of a few extreme outliers as well as high values observed particularly for Cluster 5 would make it difficult to assess differences in inter-quartile ranges and medians between clusters.

The biological meaning of the correlation between the frequency of read through transcription (downstream FPKM) and dOCR length is not clear. According to the statistics provided, these two parameters appear to be highly correlated in wild type infections and some other circumstances. However, the actual plot dispersion and the slopes show limited obvious correlation between the two parameters in many instances, with the residuals appearing to be larger than the regression. This reviewer could not find the statistical tests used to calculate the p values, or whether the statistical analyses performed evaluated whether the slope was different from 0. The statistical analyses must be fully described, and the authors may want to expand on their analyses of these correlations to better convey their potential biological meaning, in particular when they have such a large dispersion.

We now clarify in figure legends that linear regression was performed using the *lm* function in R to calculate both the slope of the regression estimate as well as the p-value for the slope being different from 0. We furthermore now address in the manuscript that transcriptional activity in the 5kb window downstream of gene 3'ends cannot completely explain the extent of dOCR induction for individual genes. Nevertheless, this analysis allowed assessing whether dOCR induction would be expected at least for some genes given the general extent of downstream transcriptional activity in a particular condition. In particular, it allowed excluding ICP0, *vhs* or ICP27 as the viral protein responsible for dOCR induction and identifying ICP22 as the responsible viral protein due to the absence of dOCR induction in Δ ICP22 infection despite strong downstream transcriptional activity in particular with PAA treatment.

In these plots, there is also a striking lack of downstream FPKM in between certain low values, which may well be artifactual. The authors should discuss the reasons for this abnormal distribution, or if it were truly artifactual, this artifact should somewhat be accounted for in the analyses.

The gaps in downstream FPKM for low values originate from the fact that at these low downstream FPKM values (< 0.1) there are only no or very few reads (0, 1, 2, 3, . . .), such that the values lacking between these low downstream FPKM values are simply not possible. Since these downstream FPKM values are also extremely noisy and ± 1 read has a massive impact on values, we have now revised these figures and analyses to include only genes with downstream FPKM values ≥ 0.05 . Notably, this increases the estimated slope of the linear regression line between downstream FPKM and dOCR length for WT and Δ ICP27 infection, reduced it slightly for Δ ICP0 and Δ vhs infection (but p-values for the slope being $\neq 0$ remained highly significant) and does not have any substantial effect for Δ ICP22 infection. The conclusions from this analysis remain the same.

The meta genome analyses of the levels of histone deposition presented in figure 4 show the discussed relative depletion past the transcription termination sites, but also present some apparent enrichments in other regions. It is unclear how these differences fit into the general model proposed.

We now also note the observed relative enrichments at gene promoters in histones and histone modification marks and particularly the increase of H3K27me3 on gene bodies. However, as we now

discuss in the manuscript, since these enrichments are also observed for genes without dOCR induction, i.e., all analyzed genes without Clusters 2, 5 and 6, they are not linked to dOCR induction. Instead, these changes can be explained by the global loss of host transcriptional activity and reduction of promoter-proximal Pol II pausing during HSV-1 infection (Birkenheuer, et al. J. Virol., 2018; Rivas et al., Mol Cell Biol., 2021).

These meta gene analyses show only genes in cluster 5, which as discussed are not representative of all genes.

We now also included metagene analyses for all protein-coding genes (**Sup. Fig. 8 and 9**). Furthermore, metagene analyses for genes without dOCR induction, i.e., all genes included in our analysis without Clusters 2,5 and 6 are also shown in **Sup. Fig. 10b-f** to distinguish changes linked to dOCR induction from other changes occurring in HSV-1 infection.

It is also unclear whether the analyses of the modified histones are corrected by histone occupancy.

Analysis of modified histones were not corrected for histone occupancy. We clarified this in the manuscript.

The analysis of the correlation between dOCR with and without induction in figure 5 only discusses the genes that have longer dOCR upon induction, omitting the genes with decreased dOCR upon induction. Moreover, no statistical analyses are performed to evaluate whether the discussed differences between genes in cluster 5 and other clusters are significant, or whether any cluster departs significantly from the average.

We presume that the reviewer refers to the changes we observed in chromatin accessibility within gene bodies and not dOCRs since we compared chromatin accessibility within gene bodies with and without induction in **Fig. 5**. Here, we previously only analyzed the 282 genes with an increase in chromatin accessibility within gene bodies (magenta points in **Fig. 5e**) and evaluated only the overlap to Cluster 5 for these genes. We have now extended this analysis to focus also on the 74 genes with a ≥ 2 -fold decrease in chromatin accessibility (purple points in **Fig. 5e**). We show that both sets of genes with either an increase or a decrease in chromatin accessibility in gene bodies upon FACT depletion in WT infection were enriched for Cluster 5 genes (Fisher's exact test, multiple testing adjusted p-value $< 10^{-5}$) but no other cluster. We hypothesize that Cluster 5 genes were preferentially affected by FACT depletion as these represent the most highly expressed genes (**Sup. Fig. 1d and 2c**) and are thus likely most dependent on FACT. Furthermore, we note that increased chromatin accessibility in gene bodies does not require poly(A) read-through and corresponding dOCR induction (e.g., **Sup. Fig. 14c**).

The authors state that "that the viral genome remains largely unchromatinized during productive infection", citing two papers presenting evidence that is consistent with this interpretation (and also with others) without discussing the many other publications using ChIP and other approaches have consistently presented evidence that it is not consistent with this conclusion. A more balanced discussion would be highly desirable.

We would like to apologize for this imprecision. We rephrased this statement to express that "the **replicating** viral genomes remain largely unchromatinized during productive infection". While the incoming viral genomes indeed become rapidly chromatinized, this replicated DNA is not as there are simply not enough free histones available. Thus, histone removal and repositioning is unlikely to be required late in infection.

The methods should be expanded to fully describe the analyses, statistics and data plotting in more detail.

We expanded the results and methods section as well as figure legend to describe the analyses, statistics and data plotting in more detail.

Reviewer #3 (Remarks to the Author):

HSV-1 disrupts transcription termination causing downstream transcription and chromatin changes to a subset of cellular genes. The viral factors and their interactions with host transcriptional mechanisms remains unresolved. This study utilizes ATAC-seq and RNA-seq/4sU-seq analysis to examine the chromatin accessibility associated with downstream transcription in primary human fibroblasts following acute infection with different HSV-1 strains and treatment with viral replication inhibitor. The results classify downstream open chromatin regions (dOCRs) that exhibit read-through transcription, and identify ICP22 and FACT as required for dOCR production. Authors conclude that ICP22 protein regulates FACT histone chaperone to alter Pol II composition and histone nucleosome repositioning.

This study provides significant insight into the chromatin and transcriptional mechanisms associated with HSV-1 infection. Overall, the technical detail is satisfactorily described and appropriately presented, and the conclusions are supported by the results.

We would like to thank the reviewer for the detailed and positive evaluation of our work.

However, there are few major critiques such as a request to provide additional data analysis for quality control as well as an analysis of the global HSV-1-induced chromatin changes beyond what is presented. In addition, there are a few minor grammatical and scientific critiques noted that should be addressed.

Supplemental ATAC-seq, Omni-ATAC-seq and ChIPmentation information and quality control should be provided. i.e., table of that provides total number of raw reads, aligned reads (both human and HSV-1), number of called peaks (ATAC-seq, ChIPmentation) and other assay-specific QC measures; promoter scores, chrM alignment rates, FRiP, etc. This analysis will support several points raised in the manuscript.

We included the respective information in new **Sup. Tab. 1** for the ATAC-seq data and **Sup. Tab. 2** for the ChIPmentation data.

For example, in lines 132-134 authors discuss PAA treatment of infected cells causing a much higher percentage of cellular reads in the ATAC-seq data due to PAA. This is unclear, are the authors referring to reads that align unique to the human genome?

We have clarified in the manuscript that cellular reads refer to reads that align to the human genome.

Are there similar read depth in the libraries that assess dOCR length differences, and how does PAA impact chrM reads in ATAC-seq experiments?

Naturally, read depth differs between samples, which is why we always also performed down-sampling of the number of cellular reads in each sample to the sample with the lowest number of reads to confirm observed dOCR length differences.

We did not observe any consistent trends in the percentage of chrM reads between conditions.

These QC results should support several statements including those from lines 134-135, 241-243, 282-283, etc. Similarly, authors conclude that dOCR represents majority of chromatin changes, however a statistical differential accessibility analysis was not performed, only dOCR length. Does the majority of differential accessibility with infection fall downstream of TTS regions (dOCRs)? Does virus infection significantly alter OCR outside of dOCR?

We clarified in the manuscript that dOCRs do not represent the only changes occurring in chromatin accessibility during HSV-1 infection. For instance, chromatin accessibility around promoters is also altered, likely due to changes in Pol II pausing during HSV-1 infection (Rivas et al., Mol Cell Biol., 2021). Furthermore, global changes in transcriptional activity and differential gene expression in HSV-1 infection also impact chromatin accessibility. However, an analysis of other changes in chromatin

accessibility during HSV-1 infection is beyond the scope of this manuscript, which focuses explicitly on changes in chromatin accessibility downstream of genes associated with disrupted transcription termination.

It is unclear how the 4,612 genes (line 125) that exhibit read-through transcription were identified. Were they determined by infection of HFF cells with one or both viral strains and with PAA treatment (i.e. all conditions)? Or were these defined by previous analysis and are representative of genes that are susceptible to this phenomenon? The description of host genes with dOCR should be more clearly described.

We have now clarified in the manuscript that these 4,162 protein-coding and lincRNA genes were identified in our previous study comparing transcriptional regulation between HSV-1 strain 17 and Δvhs infection (Friedel et al. J. Virol. 2021). These genes have no read-in transcription originating from read-through transcription of an upstream gene. We excluded genes with read-in transcription in our analysis as read-in transcription can result in dOCRs from an upstream gene extending into downstream genes (e.g., dOCRs for SRSF3 extending into downstream CDKNA1 gene, **Sup. Fig. 1a**), thus confounding dOCR analyses. No further selection regarding to whether analyzed genes are susceptible to either read-through transcription or dOCR induction was performed.

Similarly, it is unclear from results text and figure legend that among the 4,612 dOCRs, only ~500 genes are shown that 'reach at least the length indicated on the x-axis in mock and WT HSV-1 infection'. However, the minimum length indicated on the x-axis is 0 and not all 4,612 genes are shown (presumably because ~4,100 dOCRs are < ~10kb?).

We have now clarified in the figure legends that genes with a dOCR length of 0 are not shown and that **Fig. 1a** and similar figures show the number of genes on the y-axis that have dOCR length greater (not greater or equal as mistakenly stated in the original manuscript) than indicated in the x-axis, i.e., the y-axis shows the number of genes that "reach a dOCR length greater than the value indicated on the x-axis in mock and WT HSV-1 infection". For figures showing the analysis on all 4,612 analyzed genes (**Fig. 1a**, **Sup. Fig. 1c**), the y-axis was limited to 500 to better highlight the difference in the number of genes with long dOCRs between mock and HSV-1 infection. This excludes only genes with very short dOCRs, i.e. < ~5kb. We also clarified this in the figure legend. We would like to thank the reviewer for noticing this mistake.

Line 125 discusses filtering genes for read-in transcription originating from read-through transcription of an upstream gene. However, the concept of read-in is not obvious and should be defined in the background to avoid confusion.

We now define read-in transcription at the beginning of the introduction of the revised manuscript.

Also, the log₁₀ dOCR length scale of the Figure 1B doesn't seem to resolve the lengths consistent in Figure 1A. Consider changing the colors of the scale and the Figure 1B color bar and legend should specify the units of length for clarity (bp or kb?).

We have changed the colors of the scale in **Fig. 1b** to better resolve the lengths and state in the color bar that dOCR length is given in bp. Please note that we use a log scale here and not a linear scale as in **Fig. 1a** as otherwise smaller changes in dOCR length e.g., for Cluster 2 would not be visible compared to the strong dOCR induction for Cluster 5.

Line 190 suggests only one viral gene is responsible for dOCR induction.

This is of course not correct as both ICP22 and either ICP27- or stress-induced read-through transcription downstream of genes are required. We corrected the text accordingly.

Line 212 needs to reference Fig. 2c

We have added the reference to **Fig. 2c** at the corresponding place of the manuscript. Please note that this section was revised substantially, such that line numbers have changed.

Line 223-225, authors state that ICP27 is responsible for the majority of HSV-1 DoTT, but from the results presented this is not clear. Please clearly reference this data or the appropriate citation.

In collaboration with Yongsheng Shi, we previously reported that ICP27 is the main factor responsible for HSV-1-induced DoTT. We have included the respective reference (Wang et al., Nature commun. 2020). We would like to thank the reviewer for picking this up.

Authors state changes in genome-wide occupancy of histone occupancy, but the metagene analysis referenced in figure 4 is not a genome-wide analysis. An analysis of differential enrichment should be performed and statistically significant changes should be mapped with respect to gene bodies or dOCRs to support conclusions.

We have now also performed a genome-wide differential analysis for promoter regions (TSS \pm 1.5 kb), gene bodies (TSS + 1.5 kb to transcription termination site (TTS)) and downstream regions (TTS to TTS + 25 kb) for all protein-coding and lincRNA genes. We then compared distributions of log₂ fold-changes for genes in individual clusters and specific types of genomic regions (promoter, gene bodies or downstream regions) either against all other protein-coding and lincRNA genes or against the rest of the 4,162 genes included in the dOCR analysis using two-sided Wilcoxon rank sum tests. This confirmed the results from the metagene analyses showing depletion of histones and histone modifications downstream of Cluster 5 genes in HSV-1 infection with PAA treatment but not for other clusters (new **Sup. Fig. 10g-n** and **11e**). This depletion is not observed in Δ ICP22 infection (**Sup. Fig. 11f**).

It should be noted that histones and H3K27me₃ are characterized by a depletion at promoters and gene bodies but show no particular enrichment at other genomic regions that could be picked up by peak calling methods. Similarly, H3K36me₃ is only enriched in gene bodies but not downstream of genes. Thus, standard differential analyses of identified peaks as commonly done for transcription factors or other more localized histone modifications was not applicable here. Thus, we had to define windows for differential analyses in other ways as described above.

Line 271 please clarify what is meant by partly statistically significant

We have revised this sentence to state that the difference was statistically significant at some positions, i.e., bins for the metagene analysis, in the 5 kb upstream of the TTS.

Figure 2a legend should reference figure 1b and state the number of OCRs as cluster 5 alone is not clear.

We have included the reference to **Fig. 1b** and note that Cluster 5 contains 305 genes. Furthermore, we clarify that to avoid having to define a threshold on whether a particular dOCR length for a gene is considered as dOCR induction or not, we visualize dOCR lengths in each condition for all 305 Cluster 5 genes (excluding only those with a dOCR length = 0 in a particular condition) to show whether the number of genes with longer dOCRs is generally increased or not in particular conditions.

Also, it should be also clarified in the figure legend that the difference between Fig 2a and 2b is strains 17 and F, respectively. Finally, the legend states 'Experiments for (b) were performed with and without PAA treatment', but it appears PAA is in both Fig2a and 2b.

We clarified in the figure legend which strains are shown in **Fig. 2a** and **2b**, respectively and note that WT strain 17 and Δ ICP22 infection (but not the other null mutant infections) were also performed with PAA treatment for the experiment shown in **Fig. 2a**.

The methods section called 'RNA-seq controls' discusses confirmation of read-through transcription under the same experimental samples for OCR analysis, yet previously published data from alternative experiments was used to generate the figures. Supportive data indicating that read-through transcription was induced in these conditions should be provided, as well as supplemental data that indicates similar to correlations to expression in these experiments.

As also suggested by Reviewer #1, we now also include an analysis of read-through transcription, gene expression and downstream transcriptional activity using matching total RNA-seq data for the ATAC-seq experiment for mock, WT strain F and Δ ICP22 infection \pm PAA in new **Sup. Fig. 2**. Notably, scatter plots shown in **Fig. 2c,d** as well as **Sup. Fig. 6a-f** (previously **Sup. Fig. 2c-h**) already compared total RNA-seq data and ATAC-seq data from the same experiment. We only use 4sU-seq data from a previous experiment for a first initial analysis on the correlation of downstream transcriptional activity and dOCR length and to exclude that ICP0, ICP27 and *vhs* are required for dOCR induction. We now furthermore visualize matching total RNA-seq and ATAC-seq read coverage on dOCR regions for all 305 Cluster 5 genes in new **Sup. Figs. 3, 4, 5c-d, 7g-h** and **13** as well as for example genes in read coverage plots in **Fig. 3b, 5d, Sup. Fig. 1a** and **14**. Finally, we now also sequenced total RNA samples obtained from the FACT knockdown experiments that had already been obtained in parallel when performing the ATAC-seq experiments and stored at -80°C in case they were required. These are now used to compare and visualize transcription downstream of genes and chromatin accessibility for these conditions (**Sup. Figs. 13** and **14, Fig. 5d**).

Lines 59-60 states open chromatin is only observed around gene promoters and gene bodies. This is not an entirely accurate statement as accessibility can also be used to map intergenic regions.

We changed this statement to state the open chromatin is also observed in intergenic region and included a reference to the article by Thurman, et. Al. Nature, 2012, describing the landscape of accessible chromatin in the human genome.

Lines 70-72 are incomplete sentences.

We revised the respective sentences.

REVIEWER COMMENTS

Reviewer #1 (Remarks to the Author):

This is a revised manuscript that investigates how HSV-1 infection promotes the formation of open regions of chromatin that occur downstream of genes (dOCRs). The work shows that the viral protein ICP22 and the host cell factor FACT are critical for inducing dOCRs during HSV-1 infection. In this revised manuscript, the authors addressed most of my concerns, and the new data/analyses provide additional support of their model. That said, an issue remains with the data related to dOCR induction by salt/ICP22 in the absence of infection (Figure 3 and Supplementary Figure 7). I think this needs to be addressed prior to publication, as described below:

In the original experiment, Figure 3 lacked some controls. To address this, the author's performed a new experiment. In doing so, they obtained a different result regarding the impact of salt on dOCR induction. Specifically, in their first experiment salt in the absence of ICP22 did not induce dOCR formation; however, in their new experiment salt alone (i.e. - Dox) induced significant dOCR formation. The added impact of ICP22 on dOCRs in the first experiment was significant, while only a small enhancement was observed in the new experiment. The authors downplay the second result (that salt alone induces dOCRs) by attributing it to leaky ICP22 expression in absence of Dox exposure, which they state was confirmed by total RNA-seq data. There are a few issues with how the new data are considered:

1. Leaky ICP22 expression is inconsistent with their first experiment, and inconsistent with the Western blots and fluorescent images of inducible ICP22 expression shown in Supplementary Figure 7. If the authors have RNA-seq data that confirms leaky ICP22 in the new experiment, at a level significantly different than the first experiment, then this needs to be shown. The explanation that salt did not induce dOCRs in primary HFFF cells in a prior publication is not an adequate justification. In addition, the relationship between RNA-seq reads and dOCR length (e.g. the scatterplots) were not shown for the new experiment.

2. I don't think that two experiments with opposing results should be presented, especially with one in the supplement and the other in the body of the paper. If the effect of salt on dOCR formation isn't reproducible, I recommend removing it from the paper, or performing more experiments to determine what is reproducible. From my perspective, removing the salt data would have little impact on the main conclusions. The data showing that ICP22 is sufficient to induce dOCRs, as long as ICP27 is there to disrupt termination, are strong. The only point that would have to change is stating that dOCR induction is specific to disrupted termination during HSV-1 infection, which does not seem a primary take-home message of the work. Right now, the mixed result suggests it is possible for dOCR induction to occur during salt stress via a mechanism not involving viral proteins.

Minor revisions:

The labeling on many of the figures is way too small.

For plots that are restricted to genes with no known genes within 50kb downstream, please state in the legends the number of genes this includes.

Reviewer #2 (Remarks to the Author):

The revised manuscript including additional analyses and new figures and thirteen-page rebuttal letter address several of the main critiques to the original submission. Several critical issues appear not to have been addressed yet, and the new figures raise some new issues, as discussed below.

The correlation between replicates, now presented, appears limited. Figure S1b presents the correlation analyses of duplicated samples and independent treatments. In many instances, the correlation coefficient within duplicated samples is lower than between independent treatments. For example, WT str. F 8 h (1) correlation with WT str. F 8 h (2) is 0.67 but with WT str. F 17 8 h (1) is 0.71 whereas the plots in figure 1a and S1c indicate that these two treatments result in different dOCR levels.

Likewise, salt treatment of uninduced ICP22 expressing cells results in extended and frequent dOCR in the experiment presented in this submission, whereas similar levels had not been observed before in non-infected HF cells. The authors propose that “leaky” ICP22 expression may have resulted in this difference and state that the transcriptomics data show this leaky expression. These transcriptomics data are not shown, and the IF and western blot data shown in figure S7a-b fails to support significant levels of leaky expression. It is possible that salt stress itself activated ICP22 expression (different from leakiness), but these results should be shown. Were the proposed ICP22 expression leakiness model correct, then the baseline for these experiments would be biased by this leaky expression in the uninduced cells.

As an aside, the rebuttal letter argues that weaker correlation under conditions that induce no dOCR are a result of the “noise” provided by “chromatin accessibility arising from other sources.” Although not critical, it is unclear why this noise was not considered equally likely to arise from the experimental variability in the techniques.

Some results do not appear to be fully consistent the model proposed. The differences among dOCR between strains are difficult to reconcile with ICP22 playing a major role, as ICP22 is highly conserved among strains. The discussion acknowledges this conservation and ascribes the observed differences to “...subtle differences in the extent of global transcription host shut-off between these viruses” (page 16). However, the differences in dOCR across strains are not subtle and this model is testable by analyzing already available data. If the authors wish to propose this model, then they should test it.

The definition of “functional depletion” in the proposed mechanisms of action is not entirely clear and appears ambiguous. For example, the authors discuss that ICP22 recruits the FACT complex, and RNA Pol II, to the replication compartments (page 4, 11, 13) apparently as a

rationale for the work. However, the experiments include expressing ICP22 under conditions in which there are no replication compartments. It is then concluded that the sequestration to the replication compartments is not the mechanism of dOCR induction in page 15, which is fully supported by the results, only to apparently discuss again the recruitment of FACT and SPT6 to the replication compartments as a potential mechanism (page 17). Perhaps this is not the intended meaning of the second paragraph in page 17 of the introduction to the roles of ICP22 sequestering proteins into the replication compartments, but if so, the current wording is not entirely clear.

The analyses of correlation between dOCR and readthrough transcription (Figure 2c-f, Figure 3c-d, S6a-j, S7j-k) include all 4,162 genes, whereas the dOCR analyses of viral mutants that lead to the identification of ICP22 and SSRP1 as critical proteins (figure 2a, S5a-b, 7e-f, I, 12 e-f) include the 305 genes in cluster 5 only. The rationale for this difference is not clear, and one would tend to think that the same subgroup of genes should have been used for the two types of analyses. Also, as previously discussed the correlations observed in the correlation analyses appear to be driven mostly by a small fraction of “outlier” genes, whereas there appears to be no or little correlation between dOCR and readthrough transcription for most genes. Is it possible that the “outliers” are enriched in genes in cluster 5? The rationale for using different groups of genes for the different analyses should be clearly articulated, or the same subgroup of genes should be used in all analyses.

Based on published literature, the HSV1 DNA would be expected to be around 5-10% of the nuclear DNA at 8 hpi. It is unclear why the sampling of the cellular genome in PAA infections would be expected to be so much higher than in the untreated ones as to bias the analyses (page 5). As an aside, Table S1 does not appear to list the percentage of cell reads.

The analyses of readthrough transcription differences are based mostly on the 5 kb downstream the TTS, which according to Figures 1a, 2a-b, 5a-c, S1c, S5a-b, and 7e-f, appear to be OCR for a number of genes in mock infected cells. It is unclear why the 5 kb cut off was selected, or if changing this cutoff to, for example, 10 or 20 kb would affect the outcomes. It would be desirable to test the robustness of the conclusions by exploring whether changing the cut off does or does not affect the outcomes.

PAA increased dOCR lengths, which the authors ascribed to its indirect inhibition of late gene expression (page 5). However, not all late genes are equally sensitive to PAA and PAA affects DNA replication, as acknowledged in the manuscript, which recruits several potentially relevant cellular proteins to the replication compartments. It also affects the changes in nuclear architecture induced by viral replication, stalls viral DNA replication, and induces specific DNA damage responses. Although the specific mechanism is not critical for the conclusions reached, the discussion of the potential mechanisms by which PAA may increase dOCR should be more nuanced and encompassing.

The authors have slightly edited the sentence about chromatinization of viral genomes to read “replicating viral genomes remain largely unchromatinized during productive infection

(48, 49)". The addition of the word "replicating" does not address a previous critique that this sentence fails to acknowledge the very large number of publications by a number of different groups presenting abundant independent evidence of association of HSV-1 genomes with histones, or outright chromatinization, at all times after infection, or the effects of a number of epigenetic interventions in HSV-1 replication. This evidence, which is inconsistent with the statement, must at least be acknowledged.

The authors' argument that total mRNA seq data is equivalent to 4sU-seq data "As transcription downstream of genes is essentially absent prior to infection" (page 7) is not supported by data presented. Figure 1c does not show the percentage of read-through transcription in uninfected cells, and figure 1d shows a non-irrelevant level of readthrough transcripts in uninfected cells. It may perhaps be more persuasive to base the argument in the differences in read through transcription levels in cluster 5 versus all other clusters, but the % read through in mock infected cells would still be essential to support it.

Reviewer #3 (Remarks to the Author):

The authors provide a complete and thorough response to my critiques. My recommendation is to accept the manuscript in its current form.

REVIEWER COMMENTS

Reviewer #1 (Remarks to the Author):

This is a revised manuscript that investigates how HSV-1 infection promotes the formation of open regions of chromatin that occur downstream of genes (dOCRs). The work shows that the viral protein ICP22 and the host cell factor FACT are critical for inducing dOCRs during HSV-1 infection. In this revised manuscript, the authors addressed most of my concerns, and the new data/analyses provide additional support of their model. That said, an issue remains with the data related to dOCR induction by salt/ICP22 in the absence of infection (Figure 3 and Suppl. Figure 7). I think this needs to be addressed prior to publication, as described below:

In the original experiment, Figure 3 lacked some controls. To address this, the author's performed a new experiment. In doing so, they obtained a different result regarding the impact of salt on dOCR induction. Specifically, in their first experiment salt in the absence of ICP22 did not induce dOCR formation; however, in their new experiment salt alone (i.e. -Dox) induced significant dOCR formation. The added impact of ICP22 on dOCRs in the first experiment was significant, while only a small enhancement was observed in the new experiment. The authors downplay the second result (that salt alone induces dOCRs) by attributing it to leaky ICP22 expression in absence of Dox exposure, which they state was confirmed by total RNA-seq data. There are a few issues with how the new data are considered:

1. Leaky ICP22 expression is inconsistent with their first experiment, and inconsistent with the Western blots and fluorescent images of inducible ICP22 expression shown in Suppl. Figure 7. If the authors have RNA-seq data that confirms leaky ICP22 in the new experiment, at a level significantly different than the first experiment, then this needs to be shown. The explanation that salt did not induce dOCRs in primary HFFF cells in a prior publication is not an adequate justification. In addition, the relationship between RNA-seq reads and dOCR length (e.g. the scatterplots) were not shown for the new experiment.

2. I don't think that two experiments with opposing results should be presented, especially with one in the supplement and the other in the body of the paper. If the effect of salt on dOCR formation isn't reproducible, I recommend removing it from the paper, or performing more experiments to determine what is reproducible. From my perspective, removing the salt data would have little impact on the main conclusions. The data showing that ICP22 is sufficient to induce dOCRs, as long as ICP27 is there to disrupt termination, are strong. The only point that would have to change is stating that dOCR induction is specific to disrupted termination during HSV-1 infection, which does not seem a primary take-home message of the work. Right now, the mixed result suggests it is possible for dOCR induction to occur during salt stress via a mechanism not involving viral proteins.

We agree with the reviewer that the salt/ICP22 data have little impact on the main conclusions of our paper. We nevertheless tried to properly address this issue by generating new conditional ICP22 expressing HFF-tert cells, which express higher levels of ICP22 than the previously employed cells upon Dox exposure, and compared these against HFF-tert cells expressing GFP upon Dox exposure to exclude leaky ICP22 expression. We then repeated the ATAC-seq/RNA-seq experiment for both cell lines using 2h of salt stress. Unfortunately, salt stress-induced read-through transcription in this experiment was very low, i.e., barely detectable in RNA-seq, and much less than in both previous experiments. Accordingly, only traces of dOCR induction could be observed in the ICP22 expressing cells. While we would have preferred to include the salt-ICP22 data into our paper and after communicating with the editor, we thus decided to follow the recommendation of reviewer #1 and remove the salt-ICP22 data from the paper.

Minor revisions:

The labeling on many of the figures is way too small.

We revised all figures and increased font sizes of labeling.

For plots that are restricted to genes with no known genes within 50kb downstream, please state in the legends the number of genes this includes.

The number (103 genes from Cluster 5) is now included at relevant points in the manuscript.

Reviewer #2 (Remarks to the Author):

The revised manuscript including additional analyses and new figures and thirteen-page rebuttal letter address several of the main critiques to the original submission. Several critical issues appear not to have been addressed yet, and the new figures raise some new issues, as discussed below.

The correlation between replicates, now presented, appears limited. Figure S1b presents the correlation analyses of duplicated samples and independent treatments. In many instances, the correlation coefficient within duplicated samples is lower than between independent treatments. For example, WT str. F 8 h (1) correlation with WT str. F 8 h (2) is 0.67 but with WT str. F 17 8 h (1) is 0.71 whereas the plots in figure 1a and S1c indicate that these two treatments result in different dOCR levels.

It is important to note that correlation coefficients do not compare differences in the absolute levels between dOCR lengths but general trends across all genes, i.e., whether genes with longer dOCRs in one replicate/condition tend to also have longer dOCRs in the other replicate/condition. Thus, the high correlation coefficients between different virus strains and/or time-points indicate high similarity in the affected genes even if dOCR lengths were not as high, e.g., in WT str. F 8 h as in WT str. 17 8 h.

However, since the figure with the correlations led to misunderstandings, we now instead included scatter plots in Suppl. Fig. 1b,c comparing dOCR lengths either between replicates (Suppl. Fig. 1b) or between selected independent treatments (Suppl. Fig. 1c). In these figures, genes are colored according to the clusters determined in Fig. 1b. This shows that dOCR lengths for individual genes are highly reproducible between replicates, in particular for genes with large dOCR lengths (clusters 5 and 6) during HSV-1 infection with PAA treatment. In contrast, when comparing for instance WT str. F at 8 h (rep. 1) vs. WT str. 17 at 8 h (Rep. 1), dOCR lengths of cluster 5 and 6 genes are substantially higher in WT str. 17 infection than in WT str. F infection. This is consistent with results shown in Fig. 1a and b. For small dOCR lengths (i.e., in mock infection or for genes without dOCR induction), variability is much higher both between replicates and different treatments due to noise more strongly affecting small values (see response below to another comment of this reviewer). This effect is commonly also observed when comparing gene expression between replicates.

Likewise, salt treatment of uninduced ICP22 expressing cells results in extended and frequent dOCR in the experiment presented in this submission, whereas similar levels had not been observed before in non-infected HF cells. The authors propose that “leaky” ICP22 expression may have resulted in this difference and state that the transcriptomics data show this leaky expression. These transcriptomics data are not shown, and the IF and western blot data shown in figure S7a-b fails to support significant levels of leaky expression. It is possible that salt stress itself activated ICP22 expression (different from leakiness), but these results should be shown. Were the proposed ICP22 expression leakiness model correct, then the baseline for these experiments would be biased by this leaky expression in the uninduced cells.

As outlined in the response to reviewer #1 above, our independent experiment to resolve this question was unsuccessful due to limited salt-induced read-through in this experiment. After conferring with the editor, we thus decided to remove the data on salt stress from the manuscript following the recommendation by reviewer #1.

As an aside, the rebuttal letter argues that weaker correlation under conditions that induce no dOCR are a result of the “noise” provided by “chromatin accessibility arising from other sources.” Although not critical, it is unclear why this noise was not considered equally likely to arise from the experimental variability in the techniques.

We revised the manuscript to clarify that variability for genes and conditions without or with only short dOCRs is likely higher as small absolute changes due to technical and biological noise in small values (i.e., few and short dOCRs) result in large relative changes.

Some results do not appear to be fully consistent with the model proposed. The differences among dOCR between strains are difficult to reconcile with ICP22 playing a major role, as ICP22 is highly conserved among strains. The discussion acknowledges this conservation and ascribes the observed differences to “...subtle differences in the extent of global transcription host shut-off between these viruses” (page 16). However, the differences in dOCR across strains are not subtle and this model is testable by analyzing already available data. If the authors wish to propose this model, then they should test it.

We agree that it was misleading how we phrased our proposed model on page 16.

Strain 17 and F differ substantially in their virulence. This is likely due to the cumulative difference of over 300 amino acid changes in the standard set of HSV proteins [1] of both strains. In contrast to strain 17, strain F is avirulent or substantially less lethal when inoculated via the peripheral route or intracranially, respectively [2]. ICP22 is indeed highly conserved between HSV-1 strains [1]. ICP22 is perfectly conserved in its amino acid sequence between strain 17 and F. Strain KOS1.1 has six amino acid substitutions of unclear relevance compared to strain F and 17. Equally, ICP27 is also highly conserved and differs by only 2 amino acids between strains F and 17 [1]. Therefore, strain-specific differences in ICP22 or ICP27 are unlikely to explain the observed differences in dOCR induction.

We have now clarified in the manuscript that we propose a model in which the extent of dOCR induction for individual genes is determined by the absolute extent of downstream transcriptional activity in infections with different strains or conditions (8 vs 12 hpi; +/- PAA treatment). Transcriptional activity downstream of genes is not only dependent on ICP27 but on many other viral factors including the extent of HSV-1-induced host shut-off (which is substantially reduced upon PAA treatment) and an infection-induced cellular stress response. While strain-specific differences are difficult to reconcile, our model is fully consistent with the data from 8 and 12 hpi +/-PAA treatment of the different strains (Suppl. Fig. 4-6) as well as our findings from ectopic ICP22/ICP27 expression depicted in Fig. 3a,c and Suppl. Fig. 8g-f. In particular, our model is supported by the strong correlation between downstream FPKM and dOCR lengths shown in Fig. 2a,b and Suppl. Fig. 3a-c, and increased downstream FPKM values observed after PAA treatment (Suppl. Fig. 2b, 4 and 5).

We have now included a comparison of dOCR lengths between WT strains 17, F and KOS 1.1. in Suppl. Fig. 6a (all with PAA treatment). dOCRs were less induced in WT KOS 1.1 infection than in both strains F and 17. As we did not sequence total RNA from the WT strain KOS1.1 infections, which only served as a control for the respective mutant viruses on this genetic background, we cannot compare dOCR length with downstream transcriptional activity for this strain.

In summary, we feel that we provide convincing evidence that the observed strain-specific differences in the extent of dOCRs reflect differences in downstream transcription. These differences most likely result from the cumulative effects of sequence variations between the different strains in viral proteins other than ICP22 and ICP27.

The definition of “functional depletion” in the proposed mechanisms of action is not entirely clear and appears ambiguous. For example, the authors discuss that ICP22 recruits the FACT complex, and RNA Pol II, to the replication compartments (page 4, 11, 13) apparently as a rationale for the work. However, the experiments include expressing ICP22 under conditions in which there are no replication compartments. It is then concluded that the sequestration to the replication compartments is not the mechanism of dOCR induction in page 15, which is fully supported by the results, only to apparently discuss again the recruitment of FACT and SPT6 to the replication compartments as a potential mechanism (page 17). Perhaps this is not the intended meaning of the second paragraph in page 17 of the introduction to the roles of ICP22 sequestering proteins into the replication compartments, but if so, the current wording is not entirely clear.

We agree that the current wording was not entirely clear. We restructured the discussion and now state: “However, induction of dOCR upon ectopic expression of both ICP22 and ICP27 as well as our findings involving PAA treatment imply that deprivation of cellular factors by their recruitment to the viral replication compartments is not responsible for dOCR induction.”

The analyses of correlation between dOCR and read through transcription (Figure 2c-f, Figure 3c-d, S6a-j, S7j-k) include all 4,162 genes, whereas the dOCR analyses of viral mutants that lead to the identification of ICP22 and SSRP1 as critical proteins (figure 2a, S5a-b, 7e-f, I, 12 e-f) include the 305 genes in cluster 5 only. The rationale for this difference is not clear, and one would tend to think that the same subgroup of genes should have been used for the two types of analyses. Also, as previously discussed the correlations observed in the correlation analyses appear to be driven mostly by a small fraction of “outlier” genes, whereas there appears to be no or little correlation between dOCR and readthrough transcription for most genes. Is it possible that the “outliers” are enriched in genes in cluster 5? The rationale for using different groups of genes for the different analyses should be clearly articulated, or the same subgroup of genes should be used in all analyses.

We agree that the way in which we presented the correlation analysis between dOCR lengths and downstream transcriptional activity was indeed confusing. We have now restructured the manuscript to present the corresponding results for WT infection already in the section “dOCRs arise upon strong transcriptional activity downstream of genes”. This analysis serves to confirm the link between downstream transcriptional activity and dOCR induction that the analysis of downstream transcriptional activity for the identified clusters suggested. Essentially, the logic here is the following: (i) we used clustering on dOCR lengths to identify a subset of cellular genes with strong dOCR induction (i.e., Cluster 5 and to a lesser degree Clusters 2 and 6, which by definition are enriched with outliers of the total gene set used). (ii) We found that downstream transcriptional activity (i.e., downstream FPKM) mostly distinguished Clusters 5, 2 and 6 from the remaining clusters without dOCR induction. (iii) We analyzed the correlation of dOCR length and downstream FPKM to confirm this trend across all analyzed genes.

This is now clarified in the manuscript. Furthermore, we have now also included the same scatter plots with genes from Cluster 5 marked specifically to show that the “outlier” genes are indeed enriched in Cluster 5 genes. We thus focus on Cluster 5 genes for all following analyses to determine presence or absence of dOCRs in different conditions. We nevertheless also included the scatter plots comparing dOCR lengths and downstream FPKM for the mutant viruses to confirm that the same trends as for WT virus are also observed for Δ ICP0, Δ vhs and to a lesser degree Δ ICP27 infection. This confirms the results from the analysis of Cluster 5 genes that neither ICP0, vhs nor ICP27 are required for dOCR induction. In contrast, the absence of this correlation in Δ ICP22 infection despite strong transcriptional activity downstream of Cluster 5 genes (Suppl. Fig. 6c, d) confirms that ICP22 is indeed required for dOCR induction.

Based on published literature, the HSV1 DNA would be expected to be around 5-10% of the nuclear DNA at 8 hpi. It is unclear why the sampling of the cellular genome in PAA infections would be expected to be so much higher than in the untreated ones as to bias the analyses (page 5). As an aside, Table S1 does not appear to list the percentage of cell reads.

We have included a paragraph in the section 'Viral late gene expression is not required for dOCR induction' addressing the biological and technical phenomena leading to increased dOCRs in PAA-treated infected cells. Briefly, as elegantly shown by McSwiggen et al. (2019) [3], replication compartments, containing mostly uncompact DNA (and even at peak levels of viral replication much less DNA than the host cell), attract DNA binding proteins non-specifically. This will also be the case for the transposase used for ATAC-seq and OMNI-ATAC-seq. The updated OMNI-ATAC-seq protocol, in fact, addresses one major technical issue with ATAC-seq for which the uncompact mitochondrial DNA commanded much of the sequencing depth (>60%) despite only contributing to about 1% of cellular DNA. Despite dOCRs becoming more accessible in HSV-1 infection, their level of compaction is still considerably higher than that of viral replication compartments considering that viral replication compartments command a lot of the nuclear volume. Moreover, smaller amounts of viral DNA (+PAA) sequester fewer Pol II molecules away from the cellular chromatin, which substantially reduces virus-induced host shut-off. This in turn leads to substantially higher host transcriptional activity both within and downstream of genes, which leads to increased dOCRs. Together, the increased sampling capacity of the ATAC-seq enzyme (Tn5 transposase) and elevated downstream transcriptional activity lead to both better detection of dOCRs and more dOCRs, respectively. The key conclusion is that PAA treatment is ideal to detect and quantify dOCR induction as the contribution of cellular reads in the ATAC-seq libraries rises substantially (from <50% to >95%).

Table S1 now also includes the percentage of cellular reads.

The analyses of readthrough transcription differences are based mostly on the 5 kb downstream the TTS, which according to Figures 1a, 2a-b, 5a-c, S1c, S5a-b, and 7e-f, appear to be OCR for a number of genes in mock infected cells. It is unclear why the 5 kb cut off was selected, or if changing this cutoff to, for example, 10 or 20 kb would affect the outcomes. It would be desirable to test the robustness of the conclusions by exploring whether changing the cut off does or does not affect the outcomes.

It is important to note that many genes drop out of the analysis when a 10 kb or even 20 kb cut-off is chosen because of the presence of other genes within the 10 or 20 kb downstream of the respective genes. Expression of these downstream genes confounds calculation of downstream transcription, which is why we need to exclude genes with other genes on the same strand within the respective distance. We thus previously decided on the 5kb cut-off [4] as this provides a good compromise between quantifying read-through transcription in sufficiently long windows while maintaining a sufficient number of cellular genes for a comprehensive analysis. We now also calculated downstream transcriptional activity using 10 kb windows, which still includes 3,666 of the 4,162 genes from our original analysis, and correlated this to dOCR lengths for corresponding genes for all WT strains and mutants. This confirmed the original analysis as correlation coefficients between downstream FPKM in 10 kb windows. Shown below are representative data for both the 5 kb (top panels) and the 10 kb cut-offs (bottom panels) for wild-type and Δ ICP22 infection comparing dOCR length with downstream FPKM in total RNA.

Furthermore, the analyses on transcriptional activity throughout dOCR regions for cluster 5 (Suppl. Fig. 4 and 5) are not limited to the 5 kb downstream window but transcription is shown for the complete dOCR regions.

Analysis only considering the first 5 kb downstream of genes

Analysis extended to the first 10 kb downstream of genes

(data not included into the manuscript)

In summary, conclusions regarding the relationship between downstream transcriptional activity and dOCR induction are robust regarding the choice of the downstream window. This is now also mentioned in the manuscript. We thus decided not to include the figures for the 10 kb cut-offs into the manuscript.

PAA increased dOCR lengths, which the authors ascribed to its indirect inhibition of late gene expression (page 5). However, not all late genes are equally sensitive to PAA and PAA affects DNA replication, as acknowledged in the manuscript, which recruits several potentially relevant cellular proteins to the replication compartments. It also affects the changes in nuclear architecture induced by viral replication, stalls viral DNA replication, and induces specific DNA damage responses. Although the specific mechanism is not critical for the conclusions reached, the discussion of the potential mechanisms by which PAA may increase dOCR should be more nuanced and encompassing.

As outlined above in the response to another comment, we now discuss in detail in the manuscript that PAA both (i) improves detection of dOCRs as the number of cellular reads is increased, leading to improved sensitivity in detecting open chromatin regions both within and downstream of genes and (ii) leads to increased dOCRs as it reduces virus-induced host shut-off and thus leads to increased host transcription both within and downstream of genes. Furthermore, as ATAC-seq read counts gradually drop with increasing distance downstream of a respective gene, the more reads can be mapped downstream of a respective gene, the larger the region of continuous open chromatin that can be identified. PAA treatment thus increases our ability to detect longer dOCRs.

We revised the manuscript to clarify this important aspect.

The authors have slightly edited the sentence about chromatinization of viral genomes to read “replicating viral genomes remain largely unchromatinized during productive infection (48, 49)”. The addition of the word “replicating” does not address a previous critique that this sentence fails to acknowledge the very large number of publications by a number of different groups presenting abundant independent evidence of association of HSV-1 genomes with histones, or outright chromatinization, at all times after infection, or the effects of a number of epigenetic interventions in HSV-1 replication. This evidence, which is inconsistent with the statement, must at least be acknowledged.

We fully agree with the reviewer that there is ample evidence that histones are associating with the replicating viral genomes. McSwiggen et al. (2019) [3], however, showed that viral replication compartments also attract DNA binding proteins non-specifically (e.g., the lac repressor) and thus implies a certain level of openness or a state that is at least more open than cellular DNA. While the replicating viral genomes may even show outright chromatinization, it is important to note that ATAC-seq measures accessible chromatin. As is evident by the large contribution of ATAC-seq reads mapping to mitochondrial DNA, which only contributes about 1% of cellular DNA, ATAC-seq highly enriches for accessible chromatin. Even if 95% of the replicating viral genomes were to be fully chromatinized, the remaining 5% unchromatinized or only partially chromatinized viral genomes might massively contribute to the ATAC-seq libraries. While previous ChIP-PCR and ChIP-seq analysis confirmed chromatinization of the replicating viral genomes, histones will at some stage become rate limiting late in infection. Therefore, an increasing fraction of viral genomes will not be fully chromatinized late in infection.

Moreover, at late time points much of the viral chromatin does not look as one would expect from (normally) compacted DNA, which is consistent with the literature where both MNase digestion and ATAC-seq were employed (see Schang et al. 2021 for review [5]), and enables highly efficient Tn5-mediated transposition. For example, one cannot find regularly spaced nucleosomal arrays, which would lead to a more defined ATAC-seq fingerprint on viral chromatin (see also McSwiggen 2019 and Dremel & DeLuca 2019 [6]). One would also argue that read distributions in ATAC-seq data of late infected cells are, when looking at the ratio of viral to cellular DNA, unexpectedly skewed towards virus.

Nevertheless, we have now decided to delete the sentence “This may reflect the fact that the replicating viral genomes remain largely unchromatinized during productive infection [7,8] thereby alleviating the need for histone removal and repositioning during Pol II transcription of the nascent viral genomes.” We feel it is simply too speculative and not supported by sufficient data.

The authors’ argument that total mRNA seq data is equivalent to 4sU-seq data “As transcription downstream of genes is essentially absent prior to infection” (page 7) is not supported by data presented. Figure 1c does not show the percentage of read-through transcription in uninfected cells, and figure 1d shows a non-irrelevant level of readthrough transcripts in uninfected cells. It may perhaps be more persuasive to base the argument in the differences in read through transcription levels in cluster 5 versus all other clusters, but the % read through in mock infected cells would still be essential to support it.

We have now clarified in the manuscript that transcription downstream of genes is not completely absent in uninfected cells but very low compared to infected cells even in total RNA, with downstream FPKM in mock infection being 3- (without PAA) to 13-fold (with PAA) lower than in WT infection. Thus, total RNA is a cost-effective way to quantify presence of downstream transcriptional activity. Furthermore, read-through transcription is defined as the difference in the percentage of downstream

transcription (relative to gene expression) between HSV-1 infected and mock infected cells. We previously [4] defined this measure in this way because the percentage of downstream transcription was already slightly higher in mock infection (although still very low) for genes with strong read-through during HSV-1 infection than for genes without or with little read-through. Thus, % percentage of read-through in mock infected cells is set to zero by definition and was not included in the figures. We have rephrased the sentence on the definition of read-through to make this clearer.

Reviewer #3 (Remarks to the Author):

The authors provide a complete and thorough response to my critiques. My recommendation is to accept the manuscript in its current form.

We thank the reviewer for this very positive assessment of our work.

References:

1. Szpara ML, Parsons L, Enquist LW. Sequence Variability in Clinical and Laboratory Isolates of Herpes Simplex Virus 1 Reveals New Mutations. *J Virol*. 2010;84: 5303–5313. doi:10.1128/jvi.00312-10
2. Sedarati F, Stevens JG. Biological basis for virulence of three strains of Herpes simplex virus type 1. *J Gen Virol*. 1987;68: 2389–2395. doi:10.1099/0022-1317-68-9-2389
3. McSwiggen DT, Hansen AS, Teves SS, Marie-Nelly H, Hao Y, Heckert AB, et al. Evidence for DNA-mediated nuclear compartmentalization distinct from phase separation. *Elife*. 2019;8. doi:10.7554/eLife.47098
4. Hennig T, Michalski M, Rutkowski AJ, Djakovic L, Whisnant AW, Friedl M-S, et al. HSV-1-induced disruption of transcription termination resembles a cellular stress response but selectively increases chromatin accessibility downstream of genes. Kalejta RF, editor. *PLOS Pathog*. 2018;14: e1006954. doi:10.1371/journal.ppat.1006954
5. Schang LM, Hu MY, Cortes EF, Sun K. Chromatin-mediated epigenetic regulation of HSV-1 transcription as a potential target in antiviral therapy. *Antiviral Res*. 2021;192: 105103. doi:10.1016/J.ANTIVIRAL.2021.105103
6. Dremel SE, DeLuca NA. Genome replication affects transcription factor binding mediating the cascade of herpes simplex virus transcription. *Proc Natl Acad Sci U S A*. 2019;116: 3734–3739. doi:10.1073/pnas.1818463116
7. Leinbach SS, Summers WC. The Structure of Herpes Simplex Virus Type 1 DNA as Probed by Micrococcal Nuclease Digestion. *J Gen Virol*. 1980;51: 45–59. doi:10.1099/0022-1317-51-1-45
8. Dembowski JA, DeLuca NA. Selective Recruitment of Nuclear Factors to Productively Replicating Herpes Simplex Virus Genomes. Everett RD, editor. *PLOS Pathog*. 2015;11: e1004939. doi:10.1371/journal.ppat.1004939

REVIEWER COMMENTS

Reviewer #2 (Remarks to the Author):

The revised manuscript by Djakovic et al., has addressed the previous critiques. On doing so, however, an additional issue has been raised that deserves some consideration. Also, some editorial modifications are required.

Major point.

The new Supplementary figure 1b shows the variability in dOCR lengths for the clusters in which there is no dOCR induction (shades of green and blue), in that there appears to be large variability between replicated experiments in most cases (with the possible exclusion of PAA-treated infections). These data are pooled together with those from the clusters that do show dOCR induction (shades of red), which appear to show much better reproducibility between experiments. The variability within the clusters which show no dOCR induction is most likely an intrinsic limitation resulting from sampling very infrequent events, not a fault of technique or analyses. However, this variability results in several issues regarding the interpretation of the results where all genes are considered. In contrast, the intrinsic variability in the results from those particular genes does not affect any of the conclusions reached from the analyses of Cluster 5 or all clusters showing dOCR induction.

The manuscript would be strengthened if only clusters with dOCR induction were analyzed through, which is already performed in most of the critical experiments. This change will likely not affect any of the main conclusions. The authors may present a version of figure 1 separating each correlation plot in two, one including only the clusters with dOCR induction, and one including the others. It is quite likely that the correlation for the clusters with no dOCR induction will not be significant, which would provide the bases for analyzing only the clusters with dOCR induction for the rest of the paper.

Editorial issues

page 13 states, "While the increase was also observed for Cluster 5 genes (Fig 4f), it was no significant due to the relatively small number of genes in this clusters." This sentence must be edited. As written, it assumes that the differences are real and therefore will be statistically significant if the n was larger. This assumption is not supported by any analyses. With the data available, it is just impossible to conclude whether or not there was an increase for Cluster 5 genes.

PAA prevents the formation of the replication compartments, as properly discussed, but the pre-replication compartments are still formed. The discussions regarding the effects of PAA should be edited to better represent its effects.

The results obtained with histone H1 are often discussed in the context of the proposed model that ICP22 interferes with histone repositioning in the wake of RNAPII passage. Although this may well be correct, FACT and SPT6 do not participate in the exchange of H1

to the best of the knowledge of this reviewer. The effects on H1 occupancy are most likely secondary to a primary effect on the core nucleosome. The discussions pertaining H1 should be reworded.

The relevance of the discussion of the different neurovirulence of the three strains in mice to the presented results obtained in lytic infection of human primary fibroblasts is not obvious. The discussions regarding neurovirulence of these strains in mice should be removed.

The previous to last sentence of discussion regarding decimation of latent viral genomes early in reactivation does not appear to be supported, or tested, by the experiments presented in this manuscript. This speculation should be removed.

Response to the Reviewer's comments

Reviewer #2 (Remarks to the Author):

The revised manuscript by Djakovic et al., has addressed the previous critiques. On doing so, however, an additional issue has been raised that deserves some consideration. Also, some editorial modifications are required.

We are happy that the reviewer appreciated our revised manuscript.

Major point.

The new Supplementary figure 1b shows the variability in dOCR lengths for the clusters in which there is no dOCR induction (shades of green and blue), in that there appears to be large variability between replicated experiments in most cases (with the possible exclusion of PAA-treated infections). These data are pooled together with those from the clusters that do show dOCR induction (shades of red), which appear to show much better reproducibility between experiments. The variability within the clusters which show no dOCR induction is most likely an intrinsic limitation resulting from sampling very infrequent events, not a fault of technique or analyses. However, this variability results in several issues regarding the interpretation of the results where all genes are considered. In contrast, the intrinsic variability in the results from those particular genes does not affect any of the conclusions reached from the analyses of Cluster 5 or all clusters showing dOCR induction. The manuscript would be strengthened if only clusters with dOCR induction were analyzed through, which is already performed in most of the critical experiments. This change will likely not affect any of the main conclusions. The authors may present a version of figure 1 separating each correlation plot in two, one including only the clusters with dOCR induction, and one including the others. It is quite likely that the correlation for the clusters with no dOCR induction will not be significant, which would provide the bases for analyzing only the clusters with dOCR induction for the rest of the paper.

This is an excellent suggestion. We have now analyzed the reproducibility of dOCR induction separately for both genes with dOCR induction (Clusters 2, 5 and 6) and without dOCR induction (remaining clusters). The results demonstrate a substantially greater correlation for genes with dOCR induction both between replicates (Spearman rank correlation ~ 0.8 without PAA; >0.9 with PAA treatment) and between different virus strains (correlation between strain F and 17 at 8 h p.i. = 0.75 without PAA; 0.81 with PAA). We updated Supplementary Figures 1b and c accordingly. Furthermore, we moved the corresponding section in the manuscript after the point where the clusters with and without dOCR induction were identified.

Editorial issues

page 13 states, "While the increase was also observed for Cluster 5 genes (Fig 4f), it was no significant due to the relatively small number of genes in this clusters." This sentence must be edited. As written, it assumes that the differences are real and therefore will be statistically significant if the n was larger. This assumption is not supported by any analyses. With the data available, it is just impossible to conclude whether or not there was an increase for Cluster 5 genes.

We agree with the reviewer and thus deleted the second part of the sentence, namely "due to the relatively small number of genes in this clusters."

PAA prevents the formation of the replication compartments, as properly discussed, but the pre-replication compartments are still formed. The discussions regarding the effects of PAA should be edited to better represent its effects.

We agree with the reviewer that prereplication compartments are still formed in presence of PAA. However, we show that dOCR induction also occurs upon combined dox-induced expression of ICP22 and ICP27 from the cellular genome in lentivirus-transduced cells. We thus believe that differentiating between replication and prereplication compartments will only confuse the reader. Nevertheless, we have now included "large" into the respective sentence shown below to picture sequestration into large viral replication compartments.

"However, induction of dOCR upon ectopic expression of both ICP22 and ICP27 as well as our findings involving PAA treatment imply that deprivation of cellular factors by sequestration to large viral replication compartments is not responsible for dOCR induction."

The results obtained with histone H1 are often discussed in the context of the proposed model that ICP22 interferes with histone repositioning in the wake of RNAPII passage. Although this may well be correct, FACT and SPT6 do not participate in the exchange of H1 to the best of the knowledge of this reviewer. The effects on H1 occupancy are most likely secondary to a primary effect on the core nucleosome. The discussions pertaining H1 should be reworded.

We would like to thank the reviewer for pointing this out.

The SSRP1-SPT16 heterodimer (FACT) is indeed well described to bind both H2A-H2B dimers [1] and H3-H4 tetramers [2] to unwrap the nucleosomal DNA. However, SSRP1 was recently shown to also bind to the linker histone H1 as a homodimer and mediate eviction of H1 suggesting an SPT16-independent functions of SSRP1 [3]. How ICP22 induces dOCRs by interacting with SSRP1 and SPT16 thus remains unclear. Nevertheless, our results support a model in which ICP22 impairs histone repositioning in the wake of Pol II downstream of genes.

We reworded the discussion accordingly and included the respective references.

The relevance of the discussion of the different neurovirulence of the three strains in mice to the presented results obtained in lytic infection of human primary fibroblasts is not obvious. The discussions regarding neurovirulence of these strains in mice should be removed.

We agree with the reviewer and have now deleted the respective 6 lines of text.

The previous to last sentence of discussion regarding decimation of latent viral genomes early in reactivation does not appear to be supported, or tested, by the experiments presented in this manuscript. This speculation should be removed.

We deleted the respective sentence.

References:

1. Belotserkovskaya R, Oh S, Bondarenko VA, Orphanides G, Studitsky VM, Reinberg D. FACT facilitates transcription-dependent nucleosome alteration. *Science*. 2003;301: 1090–1093. doi:10.1126/science.1085703
2. Stuwe T, Hothorn M, Lejeune E, Rybin V, Bortfeld M, Scheffzek K, et al. The FACT Spt16 “peptidase” domain is a histone H3-H4 binding module. *Proc Natl Acad Sci U S A*. 2008;105: 8884–8889. doi:10.1073/pnas.0712293105
3. Falbo L, Raspelli E, Romeo F, Fiorani S, Pezzimenti F, Casagrande F, et al. SSRP1-mediated histone H1 eviction promotes replication origin assembly and accelerated development. *Nat Commun*. 2020;11: 1–15. doi:10.1038/s41467-020-15180-5